

# Cleaning up our water: reducing interferences from non-homogeneous freezing of "pure" water in droplet freezing assays of ice nucleating particles

Michael Polen[1], Thomas Brubaker[1], Joshua Somers[1], Ryan C. Sullivan[1,*]

[1]Center for Atmospheric Particle Studies, Carnegie Mellon University, Pittsburgh, Pennsylvania, USA

*Correspondence to: Ryan C. Sullivan (rsullivan@cmu.edu)

## Abstract:

Droplet freezing techniques (DFTs) have been used for half a century to measure the concentration of ice nucleating particles (INP) in the atmosphere and determine their freezing properties
to understand the effects of INPs on mixed phase clouds. The ice nucleation community has recently adopted droplet freezing assays as a commonplace experimental approach. These droplet freezing experiments are often plagued by contamination that causes non-homogeneous freezing of the "pure" water used to generate the droplets in the heterogeneous freezing temperature regime that is being measured. Interference from the early freezing of water is often overlooked and not fully reported, or
measurements are restricted to analyzing the more ice-active INPs that freeze well above the temperature of the background water. However, this avoidance is not viable for analyzing the freezing behavior of less active INP in the atmosphere that still have potentially important effects on cold-cloud microphysics. In this work we review a number of recent droplet freezing techniques showing great promise in reducing these interferences and report our own extensive series of measurements using
similar methodologies. By characterizing the performance of different substrates on which the droplets are placed and of different pure water generation techniques, we recommend best practices to reduce these interferences. We tested different substrates, water sources, droplet matrixes, and droplet sizes to provide deeper insight into what methodologies are best suited for DFTs. Approaches for analyzing droplet freezing temperature spectra and accounting and correcting for the background "pure" water
control spectrum are also presented. Finally, we propose experimental and data analysis procedures for future homogeneous and heterogeneous ice nucleation studies to promote a more uniform and reliable methodology that facilitates the ready intercomparison of ice nucleating particles measured by DFTs.

## 1 Introduction

Pure water experiences extensive supercooling. Water droplets of cloud relevant diameters (~10-
20 μm) freeze homogeneously at temperatures < -38 °C, and this temperature increases with increasing droplet volume (Koop and Murray, 2016; O and Wood, 2016). Freezing between -38 and 0 °C requires a catalyst, which in the atmosphere is provided by rare ice nucleating particles (INPs). Most precipitation over land is triggered through the ice phase (Mülmenstädt et al., 2015), and INPs may have





large impacts on cold-cloud microphysics, optical properties, lifetime, and structure (Creamean et al.,
2013; DeMott et al., 2010; Lohmann and Feichter, 2005; Yin et al., 2002).

Droplet freezing techniques (DFTs) have been utilized for decades to assess the homogeneous
freezing of pure water, and the immersion freezing properties of INPs immersed in the droplets (Bigg,
1953; Murray et al., 2012; Vali, 1971, 2014; Wex et al., 2015; Wright and Petters, 2013). In general,
these experiments work by depositing droplets containing particles onto a surface which is then cooled
down to a low temperature by a cold plate heat sink (Cziczo et al., 2017). Droplets are then assigned a
freezing temperature based on the temperature they were observed to freeze at during the cooling
process. This data is compiled to produce a plot of frozen fraction of droplets versus temperature,
referred to as the droplet freezing temperature spectrum. DFTs are utilized for both homogeneous and
heterogeneous ice nucleation experiments (Hiranuma et al., 2015; Murray et al., 2010, 2012; Vali and
Stansbury, 1966; Wilson et al., 2015; Zobrist et al., 2008). Homogeneous freezing can sometimes
present a challenge for DFTs as it is difficult to avoid interference from unintended heterogeneous
freezing (Hader et al., 2014; O'Sullivan et al., 2015; Whale et al., 2015). There are a number of variables
within DFT setups that can influence the apparent homogeneous freezing temperature of pure water
droplets that determines the background temperature spectrum and sets the lower temperature limit for
assessing heterogeneous ice nucleation. Water contamination of substrate interferences can also induce
freezing well above the homogeneous temperature limit of ~-38 °C, restricting the heterogeneous
temperature regime accessible by DFTs. Particles and cloud droplets experience a wide range of cloud
temperatures and it is important to characterize as much of the heterogeneous ice nucleation temperature
spectrum down to -38 °C as possible. This requires reducing the influence of water contaminants and
substrate effects in DFTs. Recently droplet freezing measurements in the warmer heterogeneous
temperature regime > -25 °C have been combined with measurements in the colder regime of -20 < T
< -35 °C by a continuous flow diffusion chamber to characterize the complete heterogeneous ice
nucleation temperature spectrum of ambient particles (DeMott et al., 2017). We seek to improve and
refine DFTs such that they can independently characterize the complete freezing temperature spectrum.

Nanoscale ice active surface sites on particles, macromolecules, and other surfaces are thought
to control heterogeneous ice nucleation by helping supercooled water molecules to arrange into an ice
embryo, thus reducing the nucleation energy barrier (Gurganus et al., 2014; Koop and Murray, 2016;
Marcolli et al., 2007). In DFTs the surface on which the droplets reside is thought to be one of the
biggest factors that induces non-homogeneous freezing behavior, similar to other nucleation and
crystallization processes (Diao et al., 2011; Hader et al., 2014). Properties such as the contact angle
between the droplets and the surface can be used to attempt to assess the ideality of the surface (Budke
and Koop, 2015; Koop et al., 1998; Murray et al., 2010). However, despite a large contact angle,
surfaces may have micro- or nano-scale defects that induce ice nucleation. Recent work indicates that
cracks, scratches, and other surface defects on surfaces and particles impact heterogeneous freezing
(Fitzner et al., 2015; Kiselev et al., 2016; Lo et al., 2017; Varanasi et al., 2010; Wang et al., 2016). In



general, these studies have found that defects, especially those with crystalline faces similar to ice, lower the barrier for ice nucleation and enhance ice formation above homogeneous temperatures.

Aside from surface induced effects, the environment surrounding the droplets may also influence freezing. Some research groups, including ours, deposit their droplets into an oil or other inert liquid to prevent contamination from the lab environment and eliminate the impact of the Wegener-Bergeron-Findeisen (WBF) process (Beydoun et al., 2017; Broadley et al., 2012; Polen et al., 2016; Pummer et al., 2015; Reicher et al., 2018; Wright et al., 2013; Zolles et al., 2015). The WBF process occurs when one droplet freezes and steals water vapor from unfrozen droplets, potentially inducing evaporation of nearby droplets. Contact by the growing frost halo around the frozen droplet can also induce freezing of neighboring droplets (Budke and Koop, 2015; Jung et al., 2012). Freezing assays that don't use oil typically use fast cooling rates of up to 10 °C/min so there is not enough time for these WBF effects to manifest, but this shifts the observed freezing temperature several °C colder (Mason et al., 2015). A cooling rate of 1 °C/min is more representative of typical atmospheric updraft velocities and the associated cooling rates. The oil environment prevents evaporation and these interferences, enabling slower cooling rates and droplet refreeze experiments. However, little assessment has been done to determine how or if these oils are influencing droplet freezing. We found that the surrounding squalene oil reduces the observed freezing temperature of ice active biological particles (protein aggregate macromolecules) in successive droplet freeze-thaw-refreeze experiments of Snomax bacterial ice nucleants (Polen et al., 2016). We interpreted this as caused by the hydrophobic partitioning of the largest and most ice-active macromolecules into the highly hydrophobic squalene oil that was accelerated by droplet freezing, which was previously suggested by Budke and Koop (2015). Some recent microfluidic ice nucleation techniques use fluorinated oils and/or large concentrations of surfactant to stabilize the emulsified droplets (Reicher et al., 2018; Stan et al., 2009). Their measured homogeneous freezing temperatures are typically within the expected range (-35 to -37 °C), but the surfactant may have unrecognized influences on heterogeneous freezing processes since freezing is enhanced via contact between the immersed particle and droplet interface (Durant and Shaw, 2005; Fukuta, 1975; Gurganus et al., 2014; Tabazadeh et al., 2002).

A number of non-oil immersion alternatives to DFTs have arisen in the last few years. Some groups choose to keep droplets open to air and rely on a clean, dry flow of air or $N_2$ to prevent contamination and frost growth (Whale et al., 2015). One recent study created a completely enclosed droplet chamber by sandwiching an o-ring, water, and silicon substrate between cover slips and sealing it with vacuum grease (Li et al., 2012). This resulted in a very clean environment conducive to homogeneous freezing of droplets with no need for a dry air flow over the droplets. In a comparison of droplets-in-oil and droplets-in-air, Inada et al. (2014) froze individual 3 mL droplets in n-heptane and in air and found similar freezing activity on non-coated glass slides. They correlated early freezing for these tests to the interfacial surface contact with the glass.





In addition to issues with surfaces and droplet matrixes, the "pure" water itself can introduce artifacts. Almost no work has comprehensively examined the impact of source or purity of water on homogeneous freezing. Inada et al. (2014) briefly compared tap water and MilliQ water, but these sources showed little difference when droplets were in n-heptane with a surfactant. Aside from this one report, to our knowledge, no one else has compared freezing temperatures of water from different sources. Most groups either use in-house MilliQ water systems or purchase commercial purified water, such as HPLC-grade water that is typically reserved for highly sensitive chemical analysis. A few groups additionally filter their water to remove larger particles (Hader et al., 2014; Hill et al., 2014). It is difficult to assess how well different substrates, water purification, and other method details influence the background water freezing spectrum as these important details are often not described in papers that use DFTs and the water background freezing spectrum is not always presented.

Here we report a series of experiments we have performed on the ice nucleation ability of "pure" water as is dictated by variables including the substrate, water source, and droplet matrix. The following sections describe our experimental methodology, data analysis methods, results and analysis for the aforementioned method variables, and our recommendations for best practices for future ice nucleation experiments that use DFTs. We compare our results with those of previous reports that used analogous method parameters. Finally, we advance a simple proposal for future ice nucleation experiments that will allow ready comparison between different specific measurement systems, leading to more uniform analysis that will accelerate our understanding of ice nucleation. We believe the ice nucleation community has acquired many useful strategies for dealing with issues such as contamination but that this knowledge remains largely internal within research groups and is rarely properly communicated to the larger and quickly growing community. This can discourage further advances and improvements to current designs of droplet freezing systems for INP measurements and create barriers to new groups beginning ice nucleation research. We desire to make it common practice to report these important method details and observations of pure water controls that are currently often overlooked, and begin a discussion of best practices in the community for ice nucleation experiments and droplet freezing spectrum analysis.

## 2 Droplet freezing methodology

The droplet freezing system used in this study has been updated slightly since we first described it in Polen et al. (2016). Briefly, we use an oil-immersion droplet freezing system composed of a cascade three-stage thermoelectric air-chiller (TECA, AHP-1200CAS) as the heat sink, mounted under a single-stage thermoelectric element (TE Technology Inc., VT-127-1.4-1.5-72) for fine temperature control. An aluminum sample dish sits atop an aluminum block that contains the single-stage thermoelectric element and a thermistor (TE Technology Inc., MP-3176) for temperature measurements. Our temperature measurement has an uncertainty of ±0.5 °C based on the thermistor's accuracy and our temperature calibrations. Droplets immersed in oil are placed in the aluminum dish, which is covered by a clear acrylic case for imaging by optical microscopy. No air is flown into the chamber over the oil.



Droplets are created using a variable electronic micropipette (SEOH, 3824-1LC) to deposit
droplets of 1 or 0.1 μL volume. Droplets are deposited on a substrate that sits under squalene oil (VWR,
squalene, ≥98%), mineral oil (VWR, mineral oil light), or just air. Several types of substrates were
tested in this study: hydrophobic silanized glass coverslips (Hampton Research, HR3-231), silicon
wafer chips (Ted Pella, 16007), Vaseline®, a gold wafer (Ted Pella, 16012-G), a "new" gold wafer
(Angstrom Engineering, 2WAU500-Q1), gold coated coverslips (Ted Pella, 260156-G), and solid
polydimethylsiloxane (PDMS) polymer (Dow Corning, Sylgard 184). Water for these experiments is
either from our in-house MilliQ water purifier (EMD Millipore) or bottled HPLC-grade water (Sigma
Aldrich, HPLC Plus 34877).

Substrates were cleaned or prepared in the following ways for these experiments. Silanized
cover slips were used fresh from the box without any additional cleaning. A new silanized cover slip
was used for each subsequent experiment. Silicon wafer chips were cleaned with HPLC water and
acetone and allowed to air dry before use. Gold wafer and gold cover slips were cleaned with acetone
and allowed to air dry before use. PDMS solid substrates were soaked in squalene oil for several days
before use.

A CMOS camera attached to the microscope (5x magnification) acquires an image every 5-6
seconds. We are able to view on average 40-50 1 μL droplets or 70-90 0.1 μL droplets. Frozen droplets
appear black, except in the case of a gold background in which the droplets become white. These images
are processed using a custom Matlab program that determines freezing events based on a grayscale
value (Budke and Koop, 2015; Jung et al., 2012; Reicher et al., 2018) and also determines the diameter
of each droplet. Sizing is calibrated using a 1 mm micrometer with 0.01 mm divisions. Initial tests run
on gold substrates could not be analyzed by this program because of the inverted color scale produced
by the dark gold background, so they were analyzed manually; "new" runs were analyzed using an
updated version of the program.

Data compilation and analysis is performed in one of two ways. The first is a typical statistical
analysis to determine the average and standard deviation of all runs of the droplet frozen fraction as a
function of temperature. This analysis is done when numerous arrays of many droplets have been
measured, where each array is treated as a replicate experiment. This allows us to determine standard
deviations to evaluate experiment-to-experiment variability for replicate droplet arrays. The second
approach combines all the individual arrays into a single data set. As an example, in two arrays of the
same sample type, one of the arrays had a single droplet freeze early at -25 °C and the second array had
two droplets freeze at -25 °C. In this case, combining the data would result in 3 droplets freezing at -25
°C. This second method increases the number of droplets in a set when the number of droplets is fairly
low per run; it is also used when the number of runs is small (e.g. 2 tests of a single substrate) because
statistical methods are less meaningful for low droplet counts. Figure 1 shows an example of these two
methods of data compilation of the freezing spectra. There is some deviation between the combined
unified dataset (blue) and the average of the individual replicates (red), but the combined data never




falls outside the standard deviation of the averaged data and thus we believe the combination approach is an acceptable representation of our results, especially when there are low droplet counts available for a given set of experimental parameters.

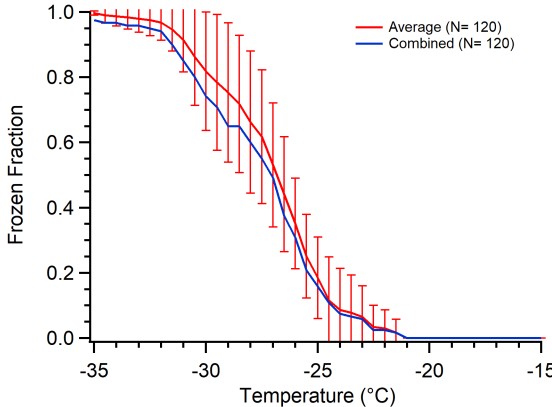

**Figure 1.** Comparison of averaging data from droplet freezing experiments on 120 total droplets measured using three replicate arrays of 40 droplets each (red with error bars) versus combining those 120 droplets into one single hypothetical array of droplets (blue). The standard deviation from the average of the three replicate arrays is shown by the vertical error bars.

## 3 Ice nucleating particle analysis

We present our data as the fraction of frozen droplets in combination with a metric derived from that freezing spectrum – the ice nuclei concentration ($c_{IN}$) – using Eq. (1) (DeMott et al., 2017; Hader et al., 2014; Hill et al., 2016; Vali, 1971, 2008). $c_{IN}$ is a droplet volume-normalized representation of the unfrozen fraction of droplets,

$$c_{IN} = -\ln(N_{unfrozen})/V_d \qquad (1)$$

where $V_d$ is the average volume of the droplets as determined by the image analysis program and $N_{unfrozen}$ is the fraction of droplets unfrozen at a given temperature. We assume the droplets are close to spherical during imaging in determining $V_d$. Hader et al. (2014) describe the derivation of Eq. (1) and present the apparent INP (or ice nuclei) concentration for pure water spectra in comparison to their particle samples. The concentration of ice nuclei per droplet volume provides a way to directly assess the impact on 200   freezing caused by a sample as compared to any contaminants or artifacts within the measurement. Normalizing the ice nuclei concentration by the surface area (or mass) of particles within the droplets defines the metric known as the ice nucleation active site density, $n_s$ (or $n_m$). $n_s$ and $n_m$ are often used in the ice nucleation literature to compare different measurements of INPs. However, there are known discrepancies when assigning $n_s$ or $n_m$ values and then comparing identical particles under widely 205   varying particle concentrations (Beydoun et al., 2016). In DFT one typically does not have any



knowledge of the contaminants that induce freezing in pure water, thus we cannot determine the density of active sites (e.g. $n_s$ or $n_m$) of the contaminants, unlike in studies of heterogeneous ice nucleation where the particle surface area or mass concentration is known or can be estimated. However, we still want to directly compare droplet freezing spectra from different experiments, and normalizing to the droplet volume provides a simple and useful way to do this. More importantly, the INP concentration is also the relevant parameter for assessing how INPs interact with and affect clouds (Hoose and Möhler, 2012). Finally, the $c_{IN}$ metric allows the ready comparison of droplet freezing spectra obtained using different droplet volumes, as different research groups use a range of droplet sizes in DFT. However, this is only possible if similar particle-in-water concentrations are used. $n_s$ or $n_m$ are often used to correct for these particle concentration differences, but as discussed these metrics may not properly account for changes caused by differing surface area or mass concentrations. The $c_{IN}$ metric, when appropriately used, is advantageous as it only assumes the INP concentration scales linearly with the droplet volume.

## 4 Results and discussion

Our results are divided into several sections that assess experimental variables tested in our DFT measurements such as substrate type and pure water generation methods. Each section begins with a brief review of previous results obtained by other ice nucleation groups using an analogous method and a discussion of why that specific method was chosen. The first section compares droplet freezing using oil immersion compared to in air. The next section goes into detail on the impact of using different sources and water purification. Then we discuss a variety of substrates examined and compare them to identify what substrates display the best performance for droplet freezing. The final section discusses tests on two droplet generation methods we used.

### 4.1 Droplet immersion matrix: oil versus air

A number of droplet freezing methods have created droplets without an oil matrix, exposing the droplets directly to air (Li et al., 2012; Mason et al., 2015; Whale et al., 2015). This method requires very clean, dry conditions to avoid artifacts such as the Wegner-Bergeron-Findeisen process and droplet contamination by aerosolized INP. In the case of the BINARY system, droplets are physically separated from one another by a PDMS mask (Budke and Koop, 2015). For systems where droplet separation is not possible, dry air or nitrogen is typically flowed over the droplets to remove ambient water vapor (Whale et al., 2015). Flowing dry air, however, exacerbates the issue of droplet evaporation and thus large droplets must be used to limit the impact of evaporation over the whole course of the temperature ramp. One unique droplet-in-air measurement was achieved by sealing a chamber completely with a single water drop deposited on the substrate in the chamber and then evaporating and re-condensing the water vapor into many smaller droplets (Li et al., 2012). This method avoids the issue of ambient water vapor altogether by turning all the sample water into vapor and re-condensing before freezing.

We have attempted droplet-in-air measurements within our own system but consistently had issues with frost halo formation upon reaching -20 °C using a standard cooling rate of 1 °C/min (Budke





and Koop, 2015; Jung et al., 2012). A series of images in Figure 2 shows this frost growth, which resulted in freezing of nearly all pure water droplets by -20 °C on hydrophobic coverslips when oil wasn't used. In comparison to other groups, our system is not air tight enough for this type of experiment. Li et al. (2012) froze their samples between two glass cover slides which were sealed together with vacuum grease for the entire experiment. Our chamber must be opened between runs which causes water vapor to condense onto the sample dish and elsewhere within the sample chamber. In this experiment, we had dry nitrogen flushing the chamber similar to previous methods but frost growth still occurred, though at much lower temperatures than tests without the nitrogen flow. Figure 2 shows the progression of frost starting at the bottom of the cover slip and continuing to grow toward the top of the glass. We consistently found that freezing and frost growth initiated around -20 °C, and we were never able to approach homogeneous freezing, likely due to our slow but realistic 1 °C/min cooling rate.

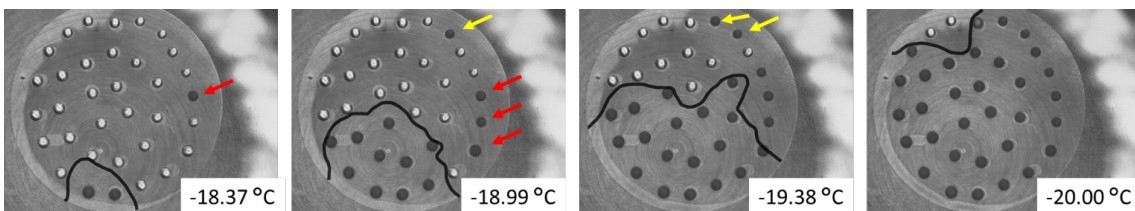

**Figure 2.** Progression of frost halos in one pure water droplet freezing experiment without an oil matrix. Dark droplets are frozen. The black line highlights the frost growth (which is visible in the image but difficult to see) spreading from the bottom left toward the top of the image. Aside from the indicated frost growth, we can also see that other droplets induce freezing in neighboring droplets, such as the droplet on the far right in image 1 (red arrow) and the top right droplet in image 2 (yellow arrow). Subsequently induced droplets are indicated by similarly colored arrows.

Many droplet freezing measurements use an oil matrix to prevent frost halos, droplet evaporation, and external contamination (Broadley et al., 2012; Pummer et al., 2015; Wright et al., 2013; Zolles et al., 2015), which is why we chose to use squalene oil for our measurements. Oil also facilitates droplet refreeze experiments to evaluate the repeatability of the ice nucleation process, and any time-dependent effects such as particle sedimentation out of the droplets (Emersic et al., 2015; Wright et al., 2013). In Polen et al. (2016), we proposed the use of mineral oil for biological samples, such as Snomax, to prevent changes in freezing behavior due to hydrophobic partitioning, which we suspected to be the case for refreezes performed in squalene oil ($C_{30}H_{50}$). However, in our attempts to use mineral oil in pure water measurements the mineral oil froze around -30 °C. We consistently saw what we at first assumed to be fogging, but upon closer inspection we found that the mineral oil had frozen completely solid, precluding droplet freezing experiments. Though we never saw mention of the freezing point in the material safety data sheets provided for the mineral oils, this is a known issue in the use of mineral oil for liquid chilling in desktop computers. However, we are also aware that the





WISDOM microfluidic DFT device uses mineral oil for droplet creation and storage (Reicher et al.,
2018). The device has successfully measured homogeneous ice nucleation down to -36 °C. Perhaps the
surfactant (Span80, 2 wt%) used to stabilize the immersed droplets prevents freezing of the mineral oil.
Alternatively, the optical fogging may not be visible when such a small volume of oil is above the
droplets, as is the case for microfluidic devices. Despite the promising results from the WISDOM
method, we are wary to suggest that any other groups attempt the use of mineral oil for droplet freezing
measurements before further investigation into how the oil's freezing may impact water droplet
freezing. For all oil-immersion experiments mentioned in the following sections, squalene oil was used
as the oil matrix, following the method of Wright and Petters (2013). Previously, we have shed light on
squalene oil reducing the observed ice nucleation activity of Snomax bacterial particles and concluded
this was due to hydrophobic partitioning of large protein aggregates (Polen et al., 2016). This was only
observed in droplet refreeze experiments of Snomax, and we do not see this effect on any other particle
sample type we have tested. Squalene oil remains our recommended immersion oil for most droplet
freezing experiments.

## 4.2 Water sources and purification

Many in the ice nucleation community use MilliQ water or similar commercial systems to purify
their laboratory's in-house water (Inada et al., 2014; Pummer et al., 2015; Rigg et al., 2013; Tobo, 2016;
Umo et al., 2015; Wright and Petters, 2013). Some groups have used bottled HPLC grade or other
similar water for their DFT (Fornea et al., 2009; Wright and Petters, 2013). Still others use alternative
methods, such as condensation, to create droplets (Campbell et al., 2015; Li et al., 2012; Mason et al.,
2015). We compared water produced by our in-house MilliQ system with bottled HPLC-grade water
from Sigma Aldrich (Figure 3). Both water types were also filtered using 0.02 µm pore size Anotop
filters before droplet generation. In general, the droplet freezing spectra obtained from the two types of
water are very similar to one another. With ~1000 droplets for each water type, we find little difference
in the apparent INP concentration as well. The biggest deviation came from the HPLC water that was
filtered for many weeks using the same Anotop filter, which shows an increase in ice nuclei around -25
300  °C, though this is not outside the standard deviation of our other samples. This result indicates that
either purchased HPLC or produced MilliQ water could be useable for droplet freezing experiments. As
MilliQ water systems use a series of filter cartridges and a membrane filter to remove dissolved
contaminants, particles, and ions from the supplied water, the quality of the produced water achieved
will depend on the quality of the original water supply source. The "house" water supply is beyond the
control of most research groups. Along with other issues we have experienced using MilliQ water that
we discuss below, high-quality bottled water may be a better and more reliable water source for ice
nucleation studies.



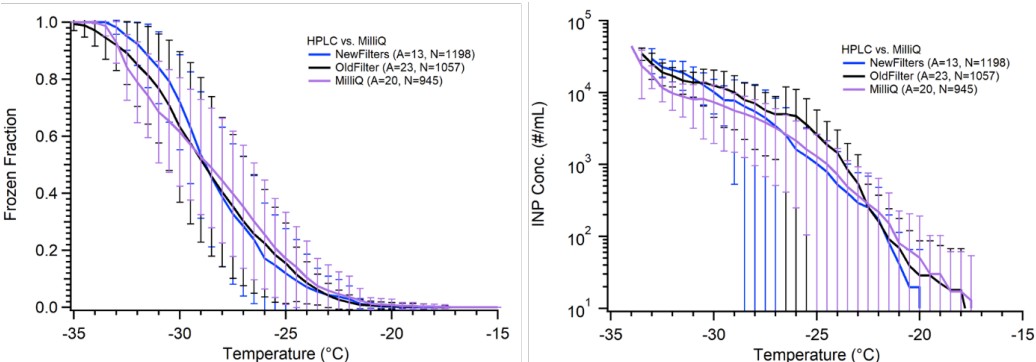

**Figure 3.** Comparison of pure water freezing using filtered MilliQ produced water and filtered purchased HPLC
water. Shown are the measured droplet freezing temperature spectra (left) and the derived INP concentration
(right). HPLC water was filtered using a new Anotop 0.02 μm filter for each bottle of water (blue), or the same
filter for multiple stock bottles of water (black). The results from typical MilliQ water arrays are shown in purple.
The parentheses next to each legend entry contains the number of arrays of droplets (A) and the total number of
droplets across all arrays (N).

We experienced significant and unexpected issues in continuing to use MilliQ water for our
droplet freezing tests and experiments that caused us to switch to bottled HPLC water for all our future
experiments. The MilliQ-produced water can result in very inconsistent results for pure water droplet
controls if the particle membrane filter is not changed on a regular basis. This is a serious concern as
there is no easy way to determine the status of the filter; the MilliQ system only measures the resistivity
of the water as a measurement of the ionic strength, as well as total carbon concentration. Figure 4
shows results from trying to diagnose the issue behind a much warmer than typical background freezing
spectrum for MilliQ water droplets. The results were highly inconsistent, with droplets in some arrays
freezing as warm as -13 °C, some droplet arrays freezing completely before -25 °C, and one array with
a median freezing temperature, $N_{50}$, of -28 °C that rivaled our least contaminated pure water tests at the
time. We also found a significant decrease in the early freezing droplets when we let the MilliQ system
run for 5 minutes before collecting water used to generate the control droplets. These caveats in using
MilliQ water will likely depend greatly on different lab environments, protocols, number of users, and
differences in the original water supply sources. Thus, we chose to perform future experiments with
bottled HPLC water in an attempt to improve experiment-to-experiment consistency by removing the
variability posed by the MilliQ system's water quality. Additionally, we filter our water before use with
a 20 nm pore size Anotop filter to further reduce variability and remove small particles that may be a
source of INPs. The use of an Anotop filter was suggested to us by Thomas Hill, as is used in the CSU
Ice Spectrometer system (Garcia et al., 2012).





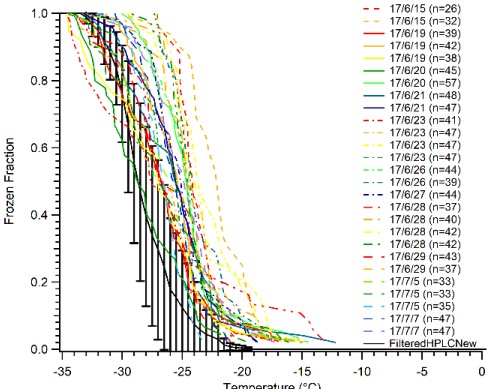

**Figure 4.** A series of tests on MilliQ generated water droplets to determine contamination sources. Droplets displayed inexplicably high freezing temperatures compared to filtered HPLC water at the time (solid brown with error bars). Temperatures for $N_{50}$ ranged from -20 °C to -29 °C from day to day. Error bars indicate standard deviation of data for the filtered HPLC water.

### 4.3 Substrate tests

In this section we discuss an extensive series of experiments in which we tested the effect of various substrates on the observed freezing spectra for pure water droplets. Our goal is to identify substrates that display a reproducibly low amount of interference in the pure water controls by allowing the droplets to freeze close to the expected homogeneous freezing temperature. This is -33 to -35 °C for the droplet volumes used here based on Eq. 7 from Pruppacher (1995). Except when noted, all arrays were created using filtered HPLC water. Each of these substrates has been shown to work reasonably well for droplet freezing experiments in the past.

#### 4.3.1 Hydrophobic cover slips

Hydrophobic cover slips are one of the most used substrates for DFTs (Bigg, 1953; Durant et al., 2008; Iannone et al., 2011; Mason et al., 2015; Murray et al., 2011; Wright and Petters, 2013). These can be made in-laboratory by silanizing a standard glass slide (Fornea et al., 2009; Wright and Petters, 2013), or can be purchased pre-silanized (Beydoun et al., 2016, 2017; Iannone et al., 2011; Mason et al., 2015; Polen et al., 2016; Whale et al., 2015; Wheeler et al., 2014). In general, results of pure water freezing on hydrophobic cover slips are variable. Whale et al. (2015) reported the 50% droplet frozen fraction ($N_{50}$) close to -26 °C for 1 μL droplets. Hader et al. (2014) reported $N_{50}$ at -30 °C for 150 nL droplets, while Iannone et al. (2011) found $N_{50}$ at -37 °C for 60 nL droplets. While an increase in homogeneous freezing temperature is expected for larger droplets, based on classical nucleation theory (CNT) we expect all of these droplet sizes to freeze homogeneously below -30 °C (Koop and Murray, 2016; Pruppacher, 1995; Vali, 1999). This implies that the larger droplets froze heterogeneously due to some unintended ice nucleating material or surface.





Our results using pre-silanized hydrophobic coverslips are similar to those reported using analogous methods by Hader et al. (2014) and Whale et al. (2015) for our larger and smaller droplets, respectively. Figure 5 displays our freezing spectra for large and small HPLC droplets on hydrophobic cover slips. The $N_{50}$ for smaller (0.1 μL) droplets (black and blue) is -29 °C, and -27 °C for larger (1.0 μL) droplets. Freezing onset begins consistently around -20 °C, and final droplets freeze between -33

to -35 °C as is expected for these droplet sizes. Importantly, we note that freezing pure water droplets simultaneously alongside sample droplets containing test particles (shown in green in Fig. 5) does not impact the freezing temperature spectrum when compared to the same droplet size data (red). This, in conjunction with the similar literature results, suggests variability between different DFT systems for pure water controls using hydrophobic coverslips may be explained primarily by the droplet size.

However, we find a counterintuitive trend when comparing the apparent INP concentration, $c_{IN}$, for these measurements. When comparing larger and smaller droplets, the concentration of ice nuclei is actually lower for larger droplets (red vs. blue points in Fig. 5). This could mean that the INP concentration for these samples is not directly related to the droplet volume but instead is more directly tied to the contact surface area with the substrate. We propose that this may be caused by one of two

effects: 1) smaller droplets have larger surface area-to-volume ratios and by normalizing to volume using $c_{IN}$ we are under-correcting interferences caused by droplet-surface contact for small droplets; or 2) larger droplets have higher contact area with the surface and thus by correcting to volume we are overcorrecting interference experienced by larger droplets. More work is necessary to connect the contact area to the elevated pure water freezing temperature. This size effect is also observed for the

gold-coated substrates discussed in Section 4.3.4.

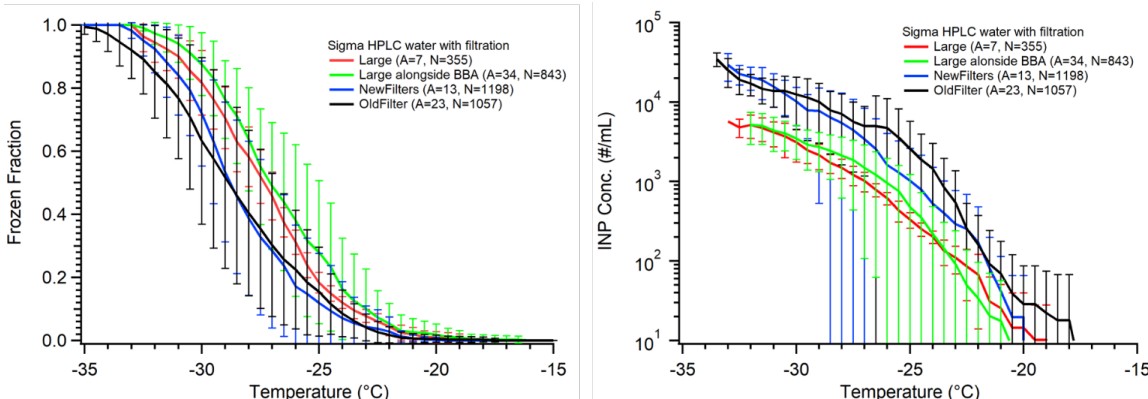

**Figure 5.** Droplet freezing temperature spectra (left) and apparent ice nuclei concentration, $c_{IN}$, (right) for pure water droplet freezing measurements on a hydrophobic cover slip. In all experiments HPLC water that was filtered using an Anotop 0.02 um syringe filter was used. Each data series has been binned into 0.5 °C temperature

increments. The red data series is from large (1.0 uL) droplets, green is from large (1.0 uL) droplets measured alongside biomass burning aerosol sample droplets (Figure 12), blue is from small (0.1 uL) droplets using a new Anotop filter for each stock bottle of filtered water, and black is small droplets using a singular Anotop filter for





many different stock bottles of water. The parentheses next to each legend entry contains the number of arrays
of droplets (A) and the total number of droplets across all arrays (N) tested for each experiment type. Error bars
are standard deviations for the replicate droplet arrays.

      We have also observed some batches of purchased coverslips to induce freezing as warm as -18
°C, and with much greater variability in the freezing spectra. Thus, it is important to evaluate each batch
of coverslips to test for these potential issues. Ideally pure water control droplets will be placed along
with droplets containing the particle sample of interest on the *same* coverslip to directly evaluate the
background freezing spectrum on that specific cover slip. This is especially important when working
with particle systems of weak ice-activity that freeze close to the background water temperature range.

### 4.3.2 Silicon wafers

      A few groups have utilized silicon wafers for droplet freezing experiments (Li et al., 2012;
Peckhaus et al., 2016). Peckhaus et al. (2016) used droplets of 107 μm in diameter and found 90% of
droplets froze below -35 °C. All droplets reported by Li et al. (2012) froze below -37.5 °C for 10-70
μm in diameter. Additionally, Li et al. performed detailed assessment of hydrophobic and hydrophilic
silicon wafers used in pure water ice nucleation experiments. They found that both types of wafer
produced nearly homogeneous freezing for pure water droplets.

      We investigated ice nucleation on silicon wafer chips typically used for SEM analysis. Several
silicon chips were placed in the sample dish with squalene oil, and 0.1 uL (~600 μm) HPLC droplets
were deposited on them. Due to the small size (5x7 mm) of the chips, the number of droplets on each
wafer chip was very low (~10), and thus we combined all the data from twelve chips as though it were
a single surface containing 120 droplets (Fig. 6). We find similar freezing activity to the hydrophobic
cover slips with onset freezing beginning around -21 °C, reaching 50% around -26 °C, and finishing at
-35 °C. The apparent INP concentration for the silicon wafer also falls close to the cover slip data (Fig.
6). We are using much larger droplets (~6-60x diameter) than the groups who have used silicon
substrates previously, so we do see higher freezing temperatures as expected. However, due to the
similar behavior and apparent INP concentration we observe using the glass cover slips and the silicon
wafer, we cannot conclude that silicon provides a more ideal surface for INP studies than silanized
hydrophobic glass. The superior performance reported by other groups using silicon wafers may be due
to higher purity water than we have access to, or other method details that make a direct comparison
challenging.





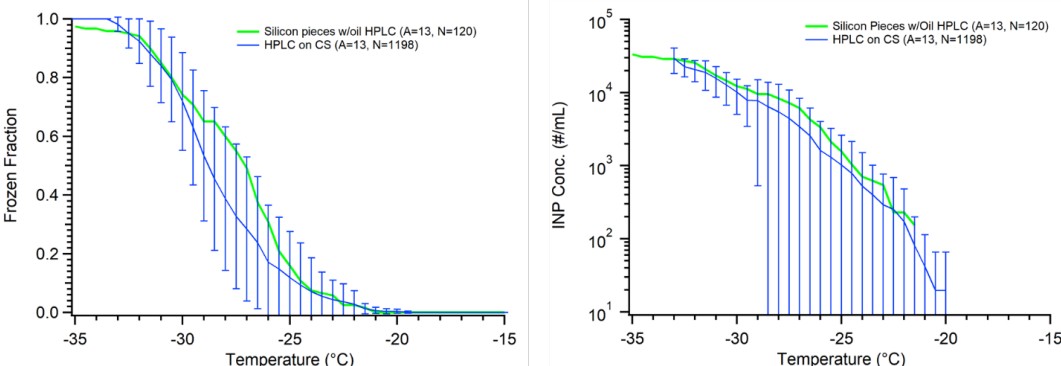

**Figure 6.** Comparison of freezing on silicon wafer chips (green) against hydrophobic cover slips (blue), following
Figure 5. The freezing temperature spectrum is on the left, and the retrieved $c_{IN}$ is on the right. Both datasets use
0.1 µL droplets. The data from all replicate arrays using silicon (green) are combined into one series and thus no
error bars can be determined. The parentheses next to each legend entry contains the number of arrays of droplets
(A) and the total number of droplets across all arrays (N).

### 4.3.3 Vaseline®

First utilized by Tobo (2016) for the Cryogenic Refrigerator Applied to Freezing Test (CRAFT)
droplet freezing instrument, Vaseline® petroleum jelly can be spread onto a clean surface to create a
makeshift hydrophobic substrate. The results from Tobo (2016) indicate great promise in this substrate
for DFT as the large, 5 µL droplets froze with $N_{50}$ = -33 °C, approaching the temperature predicted by
CNT for homogeneous freezing. We examined large (1.0 µL) droplets on Vaseline® spread onto our
sample dish in air, similar to Tobo (2016), as well as smaller droplets (0.1 µL) on Vaseline® and within
a squalene oil matrix. The results are shown in Figure 7. For tests without the oil matrix, we found quite
warm onset freezing temperatures while only a few droplets approached the homogeneous limit. We
found similar trends whether we used MilliQ water or filtered HPLC water. However, once we utilized
smaller droplets in an oil matrix, the early onset freezing vanished and we observed good background
freezing curves with lower onset and $N_{50}$ temperatures. We hypothesize that our inability to reproduce
pure water freezing near the homogeneous limit using a Vaseline® coated substrate as in Tobo (2016)
is due to the difference in cleanliness between laboratory environments as well as differences in
applying the Vaseline® layer. The oil matrix does eliminate much of the early, high temperature freezing
that is likely caused by contamination or an unevenly coated surface. This suggests the use of a laminar
flow hood or glove box may be necessary to achieve such low background freezing temperatures
without oil when the droplets are exposed to air. Tobo prepared their droplet arrays inside a glove box
within a clean room environment, and such clean conditions are not readily available to many research
groups. Uniform application of Vaseline® requires precision and a specialized spatula to get around the
lipped design, and non-uniform application will increase the risk of surface-induced freezing by any

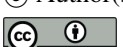



exposed underlying substrate. Interestingly, we note that one benefit to Vaseline® is we did not observe evidence of WBF effects on neighboring droplets when in air, which makes it favorable for droplets-in-air experiments if interferences can be reduced. Creation of a surface specifically designed for Vaseline® application is an important consideration if this promising technique is to be utilized more widely.

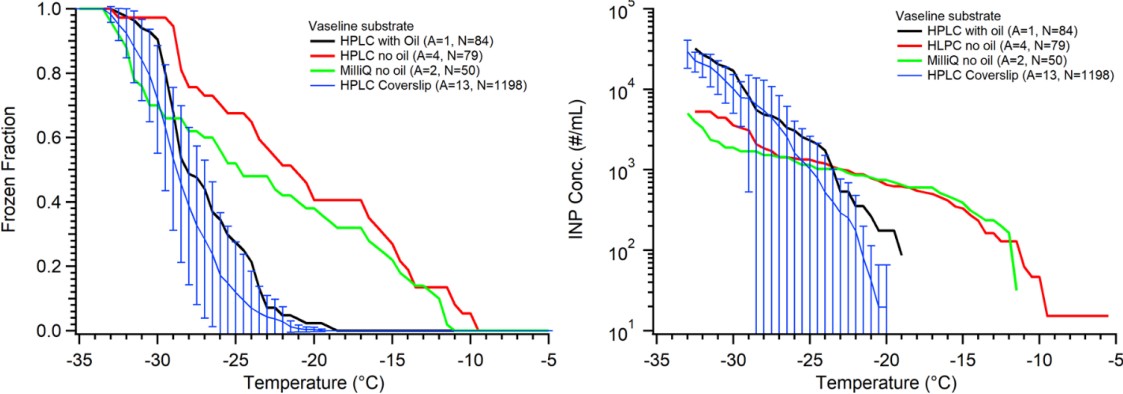

**Figure 7.** Pure water droplet experiments on a Vaseline coated substrate, following Figure 5. The HPLC water using a silanized coverslip data in blue are displayed for comparison and is the data from the hydrophobic cover slip using small droplets (Fig. 5, blue). The data from replicate arrays for Vaseline is combined as described in Section 2 and thus no error bars are determined for these. Three sets of experiments on Vaseline are shown: black is small droplets (0.1 uL) of HPLC filtered water in oil, red is large droplets (1.0 uL) without oil, and green is large droplets of MilliQ water without oil. The parentheses next to each legend entry contains the number of arrays of droplets (A) and the total number of droplets across all arrays (N).

### 4.3.4 Gold-coated substrates

Limited tests have been reported using gold-coated substrates in DFTs. Häusler et al. (2017) is the one report we are aware of that showed promise for gold as a high performance substrate in DFT, though this *Atmospheric Chemistry and Physics Discussions* manuscript was retracted during peer-review. Häusler et al. (2017) etched the surface of a gold-coated substrate and found near-homogeneous freezing temperatures ($N_{50} \approx$ -37.3 °C) for pure water droplets (45 µm) despite obvious nanoscale features in the freezing chip's cavities. In our tests we used two substrates: a gold-coated silicon wafer and a gold-coated glass cover slip (GCS). Our results are shown in Figure 8. The HPLC water on gold wafer produced a very low freezing temperature with $N_{50}$ around -32 °C; similarly small droplets of MilliQ water on the GCS had $N_{50}$ at -30.5 °C. Additionally, our first test on a second gold wafer (red) with many more droplets showed $N_{50}$ at -33.9 °C. However, large HPLC water droplets on the GCS ($N_{50}$ = -26.5 °C) froze similar to large droplets on the hydrophobic silanized cover slip ($N_{50}$ = -27 °C). When comparing the apparent INP concentrations, we again see the trend of larger droplets having lower $c_{IN}$ than smaller droplets. In this case the difference is even starker with nearly half an order of magnitude difference in $c_{IN}$ between large and small droplets on GCS at T < -30 °C. Additionally, we



find that upon cleaning and reusing a gold wafer (orange) the freezing spectrum and apparent INP concentration increased compared to the first use (red) and became similar to the silanized cover slip.

This could suggest that cleaning the surface with acetone and drying with dry, particle free air affects the surface in some way making it more ice active, or just does not adequately clean the substrate. More analysis should be performed to identify the impacts of cleaning on the gold surface. If this issue can be solved or avoided and the surface can be cleaned without introducing contamination or ice active surfaces, then gold has the potential to be a near ideal substrate. One issue with gold surfaces is they are

soft and easy to scratch, even with careful handling using Teflon-coated tweezers. This could create more ice active surface sites over time, and also be an interference in the droplet optical microscopy imaging. Gold is also much darker than the other substrates we tested, requiring manual retrieval of the droplet freezing spectrum.

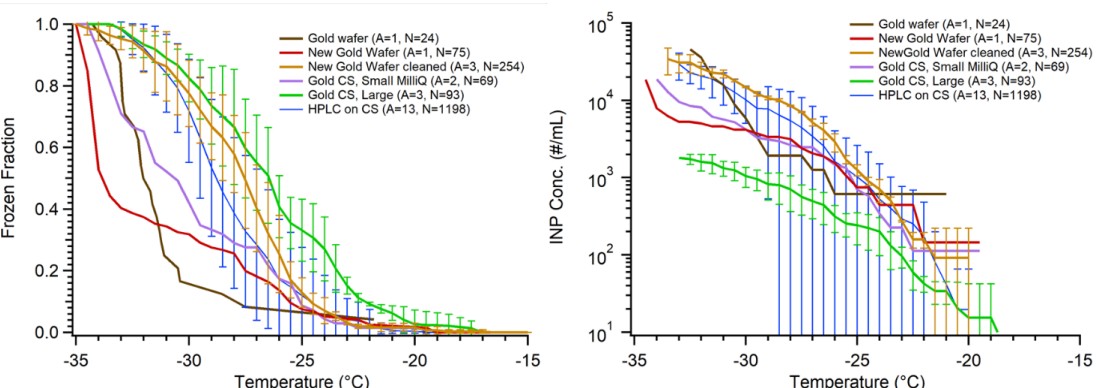

**Figure 8.** Measurements of pure water droplet freezing on gold substrates are shown following Figure 5. The data from small HPLC water droplets on a silanized cover slip are displayed in blue for comparison (Fig. 5, blue). The gold data displayed are using HPLC water droplets on a gold wafer substrate (brown), small MilliQ droplets on a gold-coated glass coverslip (GCS) (lilac), and large HPLC water droplets on a GCS (green). Also displayed are data from small droplets on another gold wafer upon first use (red), and subsequent small droplet arrays on

the same wafer following cleaning and drying, with associated error bars (orange). The parentheses next to each legend entry contains the number of arrays of droplets (A) and the total number of droplets across all arrays (N). Error bars show standard deviation from replicate droplets arrays. The data from the gold wafer (brown and red) and small droplets on a GCS (lilac) were combined into one series and so no error bars are derived.

### 4.3.5 Polydimethylsiloxane (PDMS)

Polydimethylsiloxane (PDMS) is a widely used hydrophobic, cross-linked polymeric material. PDMS has been used in microfluidic droplet freezing approaches (Reicher et al., 2018; Stan et al., 2009), but not as a substrate for conventional DFT. Reicher et al. (2018) provided a comparison of microfluidic systems with other DFTs that showed comparable homogeneous ice nucleation rates for all methods. The excellent performance of these published microfluidic techniques, and our own experience with

microfluidic devices fabricated from PDMS for DFTs led us to test PDMS as a droplet freezing



substrate. We studied two types of PDMS: a squalene oil-soaked hydrophobic PDMS surface (untreated), and a surface that was exposed to a plasma, then baked at 180 °C, and soaked in squalene oil for several days (treated). The latter represents PDMS as would be typical for a microfluidic device fabricated using conventional soft lithography. One important note is the treated PDMS did return to its
original hydrophobic form following plasma treatment and oil soaking and displayed similar freezing results as the untreated PDMS (Fig. 9). The pure water freezing spectra are again similar to our silanized cover slip results, as we have seen for most of the other substrates tested. Each of the PDMS tests was within the standard deviation of the CS data, suggesting that the PDMS surface does not provide any inherent benefit over hydrophobic silanized glass. On the other hand, PDMS is quite cheap and easy to
manipulate if you have the resources to do so, which makes it a quite useful substrate for IN studies. The hydrophobic nature of the polymer can make it prone to contamination however, and PDMS is often used as a sorbent in environmental contaminant sampling (Choi et al., 2011; Thomas et al., 2014). One other potential downside to PDMS for DFTs is its poor heat transfer properties. The thickness of the PDMS layer must be consistent for each experiment or the temperature calibration will be
inaccurate.

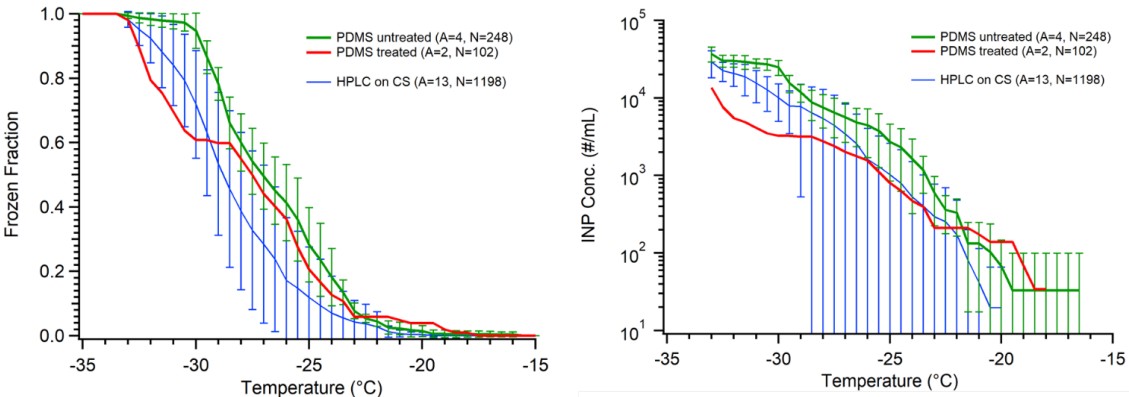

**Figure 9.** Measurements of HPLC pure water droplet freezing on PDMS are shown in red and green, following Figure 5. The data from small droplets on a silanized coverslip are displayed for comparison in blue (Fig. 5, blue). The PDMS data was obtained using treated (red) and untreated (green) PDMS polymer with small droplets. The
parentheses next to each legend entry contains the number of arrays of droplets (A) and the total number of droplets across all arrays (N). Error bars on green data show standard deviation from replicate arrays, while the red data are combined into one series as explained in section 2.

       We have recently developed a new "store-and-create" microfluidic device that shows great promise in eliminating the interferences from surface interactions as seen in our and other groups' DFTs
(Bithi and Vanapalli, 2010; Boukellal et al., 2009; Sun et al., 2011). This device will be fully described in a forthcoming manuscript. The PDMS device holds up to 600 droplets of ~6 nL volume encased in squalene oil. Each droplet is stored in an isolated microwell, completely engulfed by oil. Initial results





for pure water droplet freezing are shown in Figure 10 and compared with hydrophobic silanized cover slips. We find a $N_{50}$ around -34 °C with less than 10% of droplets freezing above -32 °C. Interestingly,
we see that the apparent INP concentration continues the same trend as the 0.1 uL droplets on a hydrophobic cover slip. This is likely because the droplets lack contact with any solid surface inside the microfluidic device and the contaminants causing this non-homogeneous freezing are related to water or oil contaminants.

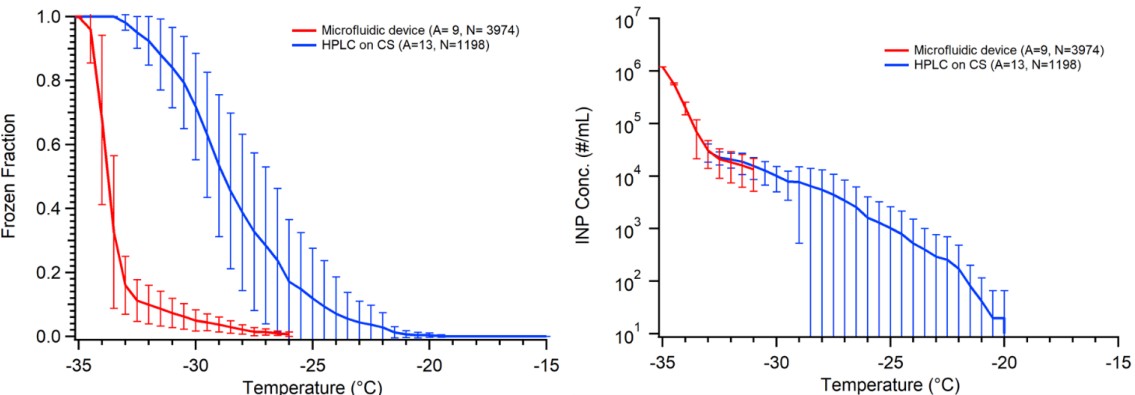

**Figure 10.** Comparison of pure water droplet freezing in our new microfluidic chip (red) versus using a silanized cover slip (CS) (blue), following Figure 5. Droplets in the microfluidic chip are 6 nL in volume and droplets on the CS are 0.1 µL. Error bars show variability of droplet freezing between different replicate arrays. The parentheses next to each legend entry contains the number of arrays of droplets (A) and the total number of droplets across all arrays (N).


### 4.4 Droplet creation methods

Some experimentation was done to compare two types of droplet creation techniques, using a syringe or autopipette. We have experienced issues with both approaches that we briefly describe here so that other users can be vigilant in avoiding these problems. Syringes create droplets with volumes of
0.1 µL that are very consistent in droplet size, much more consistent than pipettes working at similar volumes. However, using syringes has long term usage issues when the water is not completely particle free as they are difficult to clean. Each syringe (Hamilton Company, model 7001 KH) we used eventually became contaminated beyond use (evaluated by pure water control freezing spectra) and needed to be replaced. This becomes expensive when running freezing assays repeatedly for weeks and
months at a time. Syringes are also not automated and can be fragile, requiring careful use that can be time consuming when creating an array of 50+ droplets.

Switching from a syringe to an electronic pipette with disposable tips improved the long-term consistency of droplet creation. In our experience sterilized tips in boxes remain contamination free the



longest. However, we are still uncertain about the amount of contamination introduced by the pipette
tips. The best freezing experiments with pipetted droplets still freeze significantly above the
homogeneous freezing limit, which could be caused in whole or in part by pipette tips, or by remaining
water contaminants, or the silanized glass cover slip substrate.

## 5 Discussion

The results presented above provide a detailed account of many tests run on pure water ice
nucleation measurements using our cold plate DFT. Figure 11 displays a summary of the major findings
from different substrate tests. Vaseline provided the least consistency between droplet freezing
temperatures with the highest onset freezing (T= -18.5 °C), even when droplets were surrounded by oil.
However, Vaseline® had the one benefit of preventing frost-induced freezing over hydrophobic cover
slips, when droplets were not in oil. Despite this Vaseline® poses a significant number of issues, such
as uneven surface coating and unclean lab environment, which makes it impractical for many
researchers. The gold wafer showed the most promise for our standard droplet freezing method with
$N_{50}$ at -33.9 °C, but it also had some quite warm onset freezing (T= -19 °C) and when cleaned with
acetone produced a similar freezing curve to other substrates (Fig. 11). Gold wafers have the caveats
that they are quite expensive and the surface is easily scratched, as well as the potential for
contamination when cleaning, which we saw using the gold wafer ("Cleaned" vs. "New", Fig. 11).
PDMS, hydrophobic cover slips (both shown in Fig. 11), and silicon wafer chips (not shown) displayed
very similar freezing behavior with $N_{50}$ between -27 and -29 °C, only slightly warmer than the gold
wafer. Our new microfluidic device shows enormous improvements over these other methods with less
than 10% of droplets freezing warmer than -32 °C, consistently. The reason this device has such low
freezing is likely because droplets are completely engulfed by a layer of oil and have little to no contact
with the PDMS surface, unlike typical droplet-in-oil DFTs. We also observed mineral oil freezing at
temperatures warmer than homogeneous freezing, thus it should not be used for this type of analysis.
We found that MilliQ water, when the system is operating properly, displays similar ice nucleating
properties to filtered HPLC water. Few studies in the past have analyzed and compared different water
sources, so it is difficult to assess its impact on the ice nucleation results. We experienced significant
interferences using MilliQ water when the final particle filter suddenly went bad with no other
indication. This issue cost us several weeks of intensive testing to identify and resolve, which is why
we recommend the use of bottled HPLC-grade water, with additional particle filtering, to remove the
variability in the quality of the water used.






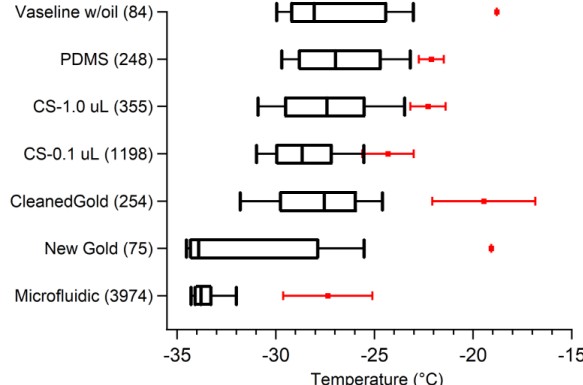

**Figure 11.** Summary of pure water droplet freezing spectra for different substrates tested. Boxes show the 25% and 75% frozen quartiles and the median, $N_{50}$, is indicated by the line inside each box. Red markers are the temperatures of the first onset freezing droplets with error bars showing variability between different replicate droplet arrays. No error bars for the onset freezing for Vaseline and gold wafer are shown because only one array was run of each. Whiskers show the 10% and 90% droplet frozen fractions. Next to each substrate name in parentheses is the number of droplets tested. Filtered HPLC water droplets produced by an electronic pipette were used in all of these measurements, except for the microfluidic chip which generated the droplets on-chip. Droplets were 0.1 μL in volume, except for the 1.0 μL on the coverslip, and the 6 nL droplets created in the microfluidic chip.

## 6. Recommendations for droplet freezing method and analysis protocols

The intent of this study is to bring to light some of the unpublished and under-reported results, experiences, and insights that are required to effectively examine heterogeneous ice nucleation using droplet freezing methods, especially when the ice nucleating particles have low freezing activity. Providing a basic overview of the best results obtained for pure water controls in our tests and the literature can lead to a series of best practices or recommendations and more method standardization. While DFTs have improved to produce accurate and reliable immersion freezing measurements, we have certainly not achieved the ideal experimental methods and strategies. To continue to advance DFTs it is important that researchers present their raw data with all its imperfections, including pure water controls, comprehensive descriptions of method details and data analysis procedures, and raw droplet freezing temperature spectra. This is the information required for the ice nucleation community to learn from each other and continue to improve our experimental methods. This will also enable new research groups to start making accurate and reproducible freezing measurements more quickly and reliably. The following are recommendations that we propose all research groups incorporate into their droplet freezing experiments and publications of these results:

1. We suggest that researchers present an assessment of raw frozen fraction curves/spectra for all types of analysis performed (homogeneous and heterogeneous freezing). This practice is often





followed in the literature, but there are plenty of instances where these data are not provided and instead the retrieved ice active site density ($n_s$, $n_m$) is the exclusive result published. Frozen fraction spectrum
is a base level analysis that all groups must do to retrieve any further parameters such as $n_s$ and $n_m$. Thus, presenting the raw frozen fraction curves for all data is a simple addition to any manuscript, even if it is presented within the supplemental section. The raw spectra can be used by the authors and others to diagnose contaminants or inconsistencies between similar droplet freezing experiments and methods.

         We encourage retrieving the apparent INP concentration, $c_{IN}$, as an especially useful metric for
quantifying the background freezing spectrum, and for comparison of different DFTs. This metric has often been used as an intermediate step to determine ice active site density, but we believe it, in and of itself, is a useful metric that should be reported, especially when examining pure water controls. Since there is no way to know the specific properties of any contaminants within pure water droplets directly, having an idea of the level of contamination per volume of water provides useful insights into what may
be preventing reaching the homogeneous freezing temperature limit. Contrary to the frozen fraction curves, INP concentration is normalized to the droplet volume, which makes it an effective way to compare pure water controls in different DFTs that invariably measure different droplet sizes.

         We will note there are some unexpected trends for our results regarding the retrieved $c_{IN}$ spectra when dealing with different droplet sizes. In particular, we see a lower concentration of ice nuclei when
we use larger droplets, despite normalizing to the volume even when the same experimental conditions are used. This suggests that normalizing to volume may over-compensate for the differences between droplet sizes. We believe this may be because the apparent INP concentration is less influenced by the concentration of particles in the water and more influenced by the contact surface area between the droplet and the surface. Thus, normalizing to volume may not be the best metric for determining activity
of contaminants in homogeneous nucleation. Fixing the droplet volume can remove this issue and is another one of our recommendations below.

         2. Procedures to correct the raw freezing spectra for interference from background freezing observed in "pure" water droplets should be reported. Retrieval of $c_{IN}$ following previous approaches (DeMott et al., 2017; Hader et al., 2014; Hill et al., 2016; Vali, 1971, 2008) and as we have done here
is our recommended approach. This background freezing spectrum should be reported, and then subtracted from the sample's spectrum. Restricting the freezing curve analysis to the 5-95% frozen droplet fraction as is now being done by some groups to exclude anomalously early and late freezing droplets is not recommended. The ice-activity of individual particles is very much a diverse spectrum, resulting in some droplets in a freezing array containing more rare ice-active INPs that induce freezing
at warmer temperatures (Augustin-Bauditz et al., 2016; Conen et al., 2011; O'Sullivan et al., 2015; Pummer et al., 2012, 2015). This can occur even in experiments on "pure" single particle type samples such as Snomax bacterial and illite NX mineral particles (Beydoun et al., 2016, 2017). Excluding the early freezing droplets would erroneously omit information on these important rare INPs whose greater ice-activity cause freezing at anomalously warm but atmospherically relevant temperatures.


3. Important method details should be documented. These include details related to the production of pure water used for droplet generation (including any additional filtration steps), any characterization of the purity of the water, and presentation of the freezing spectra for control droplets. Details regarding the substrate used and how it was prepared and cleaned are also important. Temperature calibration procedures should also be documented. DFTs are very subject to

contamination, requiring new clean surfaces and sample handling vessels to be used. This is especially a concern when working with very ice-active biological particles such as Snomax and other bacteria. Droplet preparation methods such as the pipette, syringe, or microfluidic technique used, how the particle sample was (re-)suspended in the water, and the length of time the particles spent in water prior to analysis are additional method details that may appear trivial but can have important consequences

on the observed ice nucleation properties. This is especially critical in DFT comparison studies between different groups using the same samples.

4. We recommend the use of bottled HPLC-grade or similar purchased water for droplet generation, as opposed to MilliQ-produced water. MilliQ systems can certainly produce high quality water with freezing temperatures near the homogeneous limit but are subject to sudden unannounced

changes in their water quality, and are also limited by the quality of the source water fed into the MilliQ system. Our own experiences and frustrations caused by the variability of MilliQ water has caused us to exclusively use HPLC-grade bottled water that we further filter with a 0.02 μm Anotop filter and then store in a clean glass bottle in the refrigerator.

5. Based on the findings in this study, we recommend silanized cover slips as the primary

substrate for DFT as they are the least expensive option that display the most consistent freezing behavior. Alternatively, if the cost of gold wafers is not prohibitive and measures are taken to avoid scratching the surface, then gold is a suitable substrate. Additionally, we note the incredible potential of microfluidic devices used in this study and others. We also recommend autopipettors over syringes for droplet generation due to their ease of use and reduction of potential contamination from repeated

use compared to syringes.

6. Droplet volumes and particle-in-water concentrations should be standardized as much as possible. The commonly used ice active surface site density metric ($n_s$, $n_m$) has regrettably been found to not properly normalize and correct for differences in the particle surface area or mass present in droplets during DFT. For example, just by changing particle concentration the $n_s$ values we retrieved

for illite NX shifted by several orders of magnitude (Beydoun et al., 2016). Many groups purposefully vary particle concentration to access different observable freezing temperatures, but the ice nucleation properties retrieved using different concentrations of the same system may not be consistent. The best way to evaluate this (in)consistency is to ensure overlap in the $n_s$ spectrum retrieved versus temperature, so these values can be directly compared at the same temperature. This requires using small steps in

particle concentration of about a factor of 5. Reporting the raw freezing spectra also helps to evaluate





these issues. Standardizing the total particle surface area present, by standardizing the droplet volume and particle concentration used, may also reduce these discrepancies.

7. Interferences from the substrate and/or immersion oil used, the pure water, and other potentially unrecognized sources should be regularly evaluated using pure water controls that are prepared using identical procedures as the sample droplets are. Controls should be run with a frequency determined by the level of variability in the background freezing spectrum observed using these controls, and by how close the particle sample's freezing spectrum lies compared to the background spectra. Any new batch of purchased substrates must be evaluated to assess batch-to-batch differences, which we have observed for silanized glass cover slips. Studies of low ice-activity systems such as soot particles and biomass burning aerosol require careful and extensive background control experiments. In our measurements of biomass burning aerosol we prepare a droplet array on a silanized cover slip that consists of a 1:1 ratio of pure water control droplets and BBA-containing sample droplets (Fig. 12). This provides a direct assessment of any interferences from the same substrate used for sample analysis, and equal statistics for control and sample droplets.

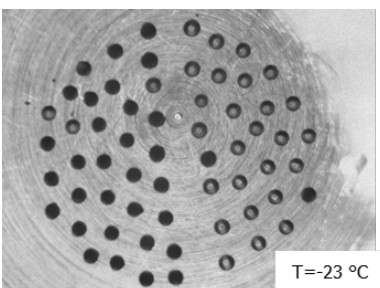

T=-23 °C

**Figure 12.** Image of droplets containing biomass burning aerosol (left half) and pure water droplets (right half) immersed in squalene oil on a silanized glass cover slip. Droplets containing aerosol sample have mostly frozen (turned dark) and pure water droplets have remained largely unfrozen (grey) at -23 °C.

8. DFTs are often evaluated by comparing measurements to published results for the same particle system. Unfortunately, we lack good reliable INP standards for proper comparison and calibration. Snomax is commonly used (Wex et al., 2015) but we identified serious issues stemming from the instability of the most ice-active ice nucleants in Snomax over time (Polen et al., 2016). This precludes Snomax as a reliable INP standard. Good comparisons have been found using illite NX minerals, but it is critical to ensure that an identical particle sample is used by each method (Hiranuma et al., 2015). Methods that collect aerosolized particles must take special care to account for their collection efficiency versus size. Just placing some material from the bulk sample into water can avoid these issues. The ice activity of mineral particles can also change with time spent in water, or by attack from strong acids. The very ice-active K-feldspar minerals are especially subject to degradation in water due to surface ion etching (Banfield and Eggleton, 1990; Holdren and Berner, 1979; Peckhaus et al., 2016). Engineered nanoparticles from inert metal oxides with reproducible particles sizes, surface





properties, and pore sizes may be the most reliable type of INP standard, though this has not yet been evaluated (Alstadt et al., 2017; Archuleta et al., 2005; Findenegg et al., 2008; Marcolli et al., 2016). Until then illite NX mineral particles are likely the best INP standard choice, provided all the above caveats are accounted for.

This study and the above series of recommendations are intended to shine light on some potential sources of inconsistencies between droplet freezing methods and create a simple, unified analysis and representation for all ice nucleation community members to follow for future publications. Many researchers already have much of the above information available before publication and use that data for detailed analysis. In the interest of moving the community forward, we seek increased transparency
regarding the aforementioned information by documenting important method details and the raw spectra for background water freezing control in all publications using droplet freezing methods.

*Author contributions*. MP and JS performed droplet freezing experiments and analysis. TB designed the microfluidic chip and analysis program and performed microfluidic device experiments. RS devised the project and recommendations for future DFT analysis. MP and RS wrote the manuscript, with input
from all co-authors.

*Competing interests*. The authors declare that they have no conflict of interest.

*Acknowledgements*. We thank Tom Hill at Colorado State University for valuable suggestions and providing the Anotop filters for our testing. Hassan Beydoun and Leif Jahn provided comments on a draft of the manuscript. This work was supported by the National Science Foundation (CHE-1554941).
MP was supported by a Graduate Research Fellowship from the National Science Foundation.

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
