# Peer review of "Cleaning up our water: reducing interferences from non-homogeneous freezing of "pure" water in droplet freezing assays of ice nucleating particles"

_Atmospheric Measurement Techniques, 2018_

## Referee Comment (RC1) · Anonymous Referee #1 · 9 May 2018

In this manuscript the authors test different substrates, water sources, droplet matrixes, and droplet sizes with the goal of improving the droplet freezing technique. Since the droplet freezing technique is being used by several groups to quantify and understand ice nucleating particles in the atmosphere, this manuscript is useful and appropriate for AMT. This manuscript will be especially useful for new researchers to the field of atmospheric ice nucleating particles. I suggest publication in AMT after the authors have had a chance to address the following comments.

1. Title. I would delete "cleaning up our water" from the title, since the manuscript

includes more than just experiments to remove impurities in water.

2. Page 9, Lines 275 – 285. Can mineral oil have different freezing temperatures depending on the average molecular weight? Maybe the WISDOM technique used a different type of mineral oil? I suggest the authors add additional information on the conditions used in WISDOM and the conditions used in the current experiments.

3. Page 9, Line 299. This sentence suggests that the water was filtered for many weeks. I assume that this is not correct.

4. Page 10, Lines 317-319 and Page 19, Lines 581-582. The issues with the MilliQ-produced water were blamed on the particle membrane filter. How do the authors know that the particle membrane filter was the source of the problem? I would have guessed any issue associated with the particle membrane filter would be rectified with the 0.02 micrometer post-filter.

5. On Page 14, it was not clear how the Vaseline surfaces were made. Line 431 suggests Vaseline was spread on the sample dish. What was the sample dish made of? On page 19, Line 563, it sounds like the Vaseline was spread on the hydrophobic glass slides? Please clarify.

6. The format of the references needs to be improved in several cases.

---

## Short Comment (SC1) · 24 May 2018

**Comment on the quantitative evaluation of background noise in drop freezing experiments**

Gabor Vali

May 24, 2018

I'd like to expand on the important topic discussed in this paper by Polen et al. The paper focuses on the factors influencing the background noise in drop freezing experiments due, among other factors, to the imperfect purity of the water used to suspend INPs for testing. In this Comment, I wish to show how the background noise can be **quantitatively** accounted for. The method is not new and, perhaps, it has been already applied by some researchers, but I am not aware of a specific exposition of it in the literature. The method is described here in some detail and its use is demonstrated with a specific example.

The paper by Polen *et al.* contains much important practical information regarding the problem of the 'background noise' unavoidably present in drop freezing experiments on heterogeneous ice nucleation[1]. Background noise in such experiments arises from several sources. The two most important and inescapable ones are (1) the fact that there is no absolutely pure water in which to suspend the INPs to be evaluated, and (2) the fact that the drops are in contact with some supporting surface. These are systematic influences which affect equally all drops in an experimental run. Additional potential systematic noise factors are: INPs in the air, other gas, or liquid covering the drops, mechanical disturbances, electrical fields, and the spread of ice from one frozen sample to another. Other items may have to be added to this list, but as far as we know now, the two main factors far outweigh the others.

Evaluation and quantitative correction for noise is the battleground of experimentalists. The first step in the battle is to keep the noise (interference) low. This is the main point addressed by Polen *et al.* as the title of their paper indicates. The second front is the evaluation and quantitative treatment of the noise. In the case of the drop freezing experiments, the solution to this second problem is approached by the use of background tests, a point well emphasized in the paper. To resume briefly, in situations where the
* * *
[1] The generic description as 'drop' freezing is applied in the paper and in this comment to stand for any manner in which a bulk sample is divided in numerous small sub-samples in order to observe the distribution of INPs of various activity. The resulting distribution is called the spectrum of INP activity.

[Figure]

Figure 1: Fraction of sample drops and control drops frozen as a function of temperature.

sample is a laboratory preparation, i.e. the material to be tested is added to purified water, the background can be evaluated by testing the purified water in exactly the same manner as the INP-bearing sample is. This can be done either in separate tests, or by simultaneously observing drops of the purified water and those already containing the sample in the same run. For testing for the INP content of water samples like rain, snow, etc., determination of the background consists of parallel tests with the most highly purified water available[2]. This provides an assessment of the noise level arising from the supporting surface(s) and other factors, and it is assumed that the measured INP content exceeding this level is a true part of the sample. Non-systematic effects, like frost spreading on the supporting surface, have to eliminated because they cannot be corrected for with noise subtraction.

As already implied in the foregoing, the basic assumption in drop freezing experiments is that the observed INP content is the sum of the background and of the sample. There are limits to the validity of this assumption, such as possible dissolution of some potential INPs in the sample of solid material, but this possibility is covered by referring to the tests as probing for INPs active in "immersion freezing". In general, the very nature of the drop freezing tests is based on the additive assumption, since INPs are counted and the results are given as normalized values of the number concentration or total surface area of INPs.

Consider the results shown in Figure 1 for an experiment with a sample of soil suspension and with a control sample with the distilled water. The same drop volumes were
* * *
[2]It is unlikely to be necessary, but it may be useful to simulate the ionic composition of the rain or snow sample just in case the sample to be tested is has some chemical properties that may have some interaction with the supporting surface.

used for both. As can be seen, there is substantial overlap between the temperature ranges of the obseerved freezing events for the sample and for the control. This appears to be alarming as an indication of lack of purity of the distilled water, i.e. a high background INP content. This may be thought to invalidate the portion of the data in the region of overlap, or the temptation may arise to subtract the fraction frozen for the control from the fraction frozen for the sample. This would be an error.

A simple and direct subtraction of the noise level from the signal is available in terms of the differential nucleus spectra defined in Vali (1971, V71). Time-dependence is not considered and is a minor factor in any case (cf. V71 and numerous recent publications). With the sample and the control data taken with the same cooling rate the importance of time is further reduced. The differential spectra offer a clear and intuitive way of achieving the noise subtraction whether it is done in terms of number concentration or surface area. These spectra express the number of INPs per unit volume of water, or per particle surface area. For simplicity, the example is presented here in terms of the number of INPs per unit volume per temperature interval. As given by Eq. 1 in V71:

$$k(T) = -1/(V\Delta T) * ln(1 - \Delta T/N(T))$$

where $T$ stands for temperature in $^{\circ}$C, $N$ is the number of drops not frozen and $V$ is the volume of the drops. It is to be remembered that this expression is the result of considering that a freezing event in the interval $\Delta T$ is the result of a drop containing *at least* one INP active in that temperature interval. For relatively small $\Delta T$-values and for large $N$ this approximation to having a *single* INP per drop responsible for the observed freezing event is very good (and can be quantified). Under these circustances it is entirely appropriate to take $k_{\text{corrected}}(T) = k_{\text{sample}}(T) - k_{\text{control}}(T)$.

For the data shown in Fig.1, the results in terms of $k(T)$ are shown in Fig. 2. As can be seen the actual correction is small over most of the temperature range of the sample except in the region of the dip of the spectrum where the ratio $k_{sample}/k_{control}$ becomes small, almost becoming equal near -17$^{\circ}$C. The corrected value is lower than the control in that region. Here, acceptance of the corrected value may well be questioned and the question would have to be examined considering the probable errors in both the the soil sample and the control in terms of the sample sizes involved, namely the $\Delta N_i$ values for each $\Delta T_i$ in the region of interest. In fact, that evaluation should be done for all parts of the spectrum but for the majority of points in this example the significance of the corrected spectrum appears to be assured by the small values of the corrections. For the sake of brevity, this statistical evaluation is not entered into here. It is worth noting that the relatively minor correction seen in Fig. 2 is in contrast with the impression given by Fig. 1 for a possibly more important impact of the background due to the distilled water used.

The differential spectrum is used in the foregoing discussion because it is the most straightforward for the purpose, specially in the potential to evaluate statistical errors

[Figure]

Figure 2: Differential spectra for the two data sets shown in Fig. 1. The line labelled 'corrected' represents $k_{\text{corrected}}$.

for each temperature interval. The cumulative spectra, $K(T)$ in V71, is an integral of the differential spectrum and correction for background noise (distilled water INPs) can be also made in terms of K(T). It may be noted that in Section 3 of the paper under discussion the cumulative spectra are introduced as $c_{IN}$ in their Eq. 1, with $N_{\text{unfrozen}}$ designating the fraction, not the number of unfrozen drops as in this comment. This equation is the same as Eq. 13 in V71.

In summary, measurements of INP content by drop-freezing experiments can be evaluated with quantitative corrections for the INPs that may be introduced with the water carrying the sample to be tested, or whatever other systematic factor contributes to the background freezing events. Such corrections increase confidence in the results, and in extreme cases indicate when the results cannot be trusted due to high background levels. If needed, statistical confidence levels can be computed. It should be noted that the ultimate reliability of results derived from drop freezing experments is determoned in many cases – above and beyond the corrections for background influences – by difficult to control time-varying variables. Time varying factors are, for example, aging of the sample itself, problems of controlling particle sizes precisely from experiment to experiment, the settling of particles in the storage containers, and more. These problems nonewithstanding, drop freezing experiments have many uses; the paper by Polen *et al.*, as well as this comment, may help to further increase scientific gains from these experiments.

---

## Short Comment (SC2) · 30 May 2018

In the meantime the article Häusler, T., Witek, L., Felgitsch, L., Hitzenberger, R. and Grothe, H.: Heterogeneous freezing of super cooled water droplets in micrometre range - freezing on a chip, Atmos. Chem. Phys. Discuss., (January), 1–19, doi:10.5194/acp-2017-31, 2017 has been published in Häusler, T.; Witek, L.; Felgitsch, L.; Hitzenberger, R.; Grothe, H. Freezing on a Chip - A New Approach to Determine Heterogeneous Ice Nucleation of Micrometer-Sized Water Droplets. Atmosphere 2018, 9(4), 140; https://doi.org/10.3390/atmos9040140

---

## Referee Comment (RC2) · B. Murray (Referee) · 9 Jul 2018

In this manuscript Polen et al. describe a study of non-homogeneous ice nucleation in 'pure' water samples and a variety of substrates. They then go on to make a sequence of suggestions for groups using droplet freezing assays. This is a highly valuable manuscript and it will help both newcomers to the field and established groups improve their droplet freezing techniques. I agree with the authors statement that individual groups have a great deal of knowledge on this subject, which isn't necessarily made publically available. The comments I have made below are made in this spirit. I support the manuscript's publication and list a few comments which I hope the authors

will use to further strengthen their manuscript:

1.) Handling blank experiments vs. 'pure' water control experiments. In the list of recommendations, I encourage the authors to include a recommendation that experimentalists conduct full handling blank experiments in addition to 'pure' water control experiments. By handling blank experiment I mean putting the water used in the experiment through the full process that the water containing a sample had been through. In Vergara-Temprado et al. (1) we demonstrated that this was critical. The 'pure' water control experiments were lower than the handling blanks (see Figure 2). The experiments where droplets were loaded with black carbon froze at a similar temperature to the handling blanks, but were above the pure water control. We consequently reported limiting values. In Fig 12, is the control experiment a handling blank?

2.) Abstract: The word 'plagued' is perhaps a bit strong. It is a limitation.

3.) The authors refer to homogeneous nucleation at -38 C and the 'homogeneous nucleation limit'. This creates the impression that homogeneous nucleation has a well-defined limit where it occurs. It is volume and time dependent and this matters. For example, a 1 um droplet cooled at 1 K min-1 will have a freezing probability of 0.5 at -33.5C (according to Koop and Murray (2)). Furthermore Herbert et al.(3) found that enough cloud droplets started to freeze in a cloud to start to affect cloud properties at around -33 C or so (depending on the homogeneous parameterisation), even though 50% of droplets would only freeze homogeneously at around -38 C. I would like to see the introductory sentences adjusted to be less definitive about the when homogeneous nucleation becomes important. Also, at ln 51, define the 'homogeneous nucleation limit' as, for example, the T at which 50% of 10 um droplets are expected to freeze on cooling at 1 K min-1. 4.) Ln 34-35. Vergara-Temprado et al. (4) could be cited here, this paper clearly shows a sensitivity of clouds to INP. 5.) Ln78. Replace 'steals' is not the best choice of words. 6.) Ln93-97: Tarn et al. (5) also used microfluidic technology to study heterogeneous freezing. One of the objectives was to see if the oil and surfactant influenced nucleation. They measured ice nucleation (ns) for a range of

materials and compared to literature. The results suggest that these technologies can be used to make droplets and study heterogeneous nucleation.

7.) Ln 214. 'correct for', replace with 'account for'. This isn't a correction.

8.) Fig 2 and discussion. These images are similar to those shown in Fig 4 or Whale et al. (6), mention that the new findings are consistent with what was previously found. Also, note that it is relatively easy to see if this is a problem.

Also, just for this discussion: we have found that this becomes a more significant problem when doing experiments in a humid environment. We solved this problem by improving the design of our ul-NIPI chamber to make it more air tight.

9.) 'Particle sedimentation out of the droplet'. I don't think this is very likely. They may sediment to the bottom of a droplet, but are unlikely to sediment across an interface. Emersic et al. provide no evidence that particles can sediment out of a droplet.

10.) P 9-10. We have also compared HPLC water and compared to water from MiliQ systems. We have found that the quality of HPLC water is also variable. I would recommend that whatever the source of water, the experimenter should demonstrate its quality and do the experiments to test the quality at sufficient intervals. I also agree that water from MiliQ machines can be highly variable, but in our experiments if the machine is well-maintained then the quality is systematically high, although we too have had periods when the quality was much lower.

11.) Fraction frozen plots throughout. I think it would be valuable to show the theoretical homogeneous nucleation curve. I would suggest using Koop and Murray (2) since in this paper the authors attempted to constrain classical theory in a physically plausible way, which gives more confidence in the values at higher temperatures (relevant for ul sized droplets) where there is very little or no data.

12.) Substrate dependent nucleation: Mention the result of Price et al. (7) (Figure 4) where it was shown that a Teflon substrate produced lower freezing temperatures

when compared to a salinized glass surface. 13.) Ln 713: On the topic of K-feldspar being sensitive to water. We explored this in Harrison et al. (8) and showed that BCS 376 (a standard feldspar which is available for anyone to buy) only degraded by ∼1 C over 16 months in water. Other feldspars, in particular those exhibiting hyperactivity, are more sensitive to time in water.

14.) Also, I think Vali's comment on this paper is really valuable, and it would be useful to record this approach in the literature (either in Polen et al. or Vali could consider a short not in AMT setting this out formally?).

References: 1. Vergara-Temprado J, et al. (2018) Is Black Carbon an Unimportant Ice-Nucleating Particle in Mixed-Phase Clouds? J. Geophys. Res. 123(8):4273-4283. 2. Koop T & Murray BJ (2016) A physically constrained classical description of the homogeneous nucleation of ice in water. J Chem. Phys. 145(21):211915. 3. Herbert RJ, Murray BJ, Dobbie SJ, & Koop T (2015) Sensitivity of liquid clouds to homogenous freezing parameterizations. Geophys. Res. Lett. 42(5):1599-1605. 4. Vergara-Temprado J, et al. (2018) Strong control of Southern Ocean cloud reflectivity by ice-nucleating particles. P. Natl. Acad. Sci. USA. 5. Tarn MD, et al. (2018) The study of atmospheric ice-nucleating particles via microfluidically generated droplets. Microfluid. Nanofluid. 22(5):52. 6. Whale TF, et al. (2015) A technique for quantifying heterogeneous ice nucleation in microlitre supercooled water droplets. Atmos. Meas. Tech. 8(6):2437-2447. 7. Price HC, et al. (2018) Atmospheric Ice‐Nucleating Particles in the Dusty Tropical Atlantic. J. Geophys. Res. 123(4):2175-2193. 8. Harrison AD, et al. (2016) Not all feldspars are equal: a survey of ice nucleating properties across the feldspar group of minerals. Atmos. Chem. Phys. 16(17):10927-10940.

---

## Author Comment (AC1) · 6 Aug 2018

Thank you for providing the new reference. We have updated the citation in the revised manuscript.

Regards, Ryan Sullivan
* * *

---

## Author Comment (AC2) · 6 Aug 2018

**Response to Referee #1**

In this manuscript the authors test different substrates, water sources, droplet matrixes, and droplet sizes with the goal of improving the droplet freezing technique. Since the droplet freezing technique is being used by several groups to quantify and understand ice nucleating particles in the atmosphere, this manuscript is useful and appropriate for AMT. This manuscript will be especially useful for new researchers to the field of atmospheric ice nucleating particles. I suggest publication in AMT after the authors have had a chance to address the following comments.

**We thank the referee for their positive response to our manuscript. We have responded to each comment made in turn along with the revision made to the manuscript where appropriate.**

Title. I would delete "cleaning up our water" from the title, since the manuscript includes more than just experiments to remove impurities in water.

**While we understand the referee's suggestion, we prefer to retain the original title. The freezing artifacts caused by substrate interactions can appear to have a similar and difficult to distinguish effect on the background freezing temperature in "pure" water controls, and thus we feel our manuscript does largely focus on the effects of real and apparent water impurities. Gabor Vali made similar a similar assessment in his comment on the discussion paper. Furthermore, if substrate interference can be eliminated (such as in our microfluidic approach), then any remaining background freezing above the expected homogeneous freezing temperature are due to any trace and difficult to eliminate water impurities. We think the title will make the focus and purpose of our manuscript clearer and thus reach a wider audience.**

Page 9, Lines 275 – 285. Can mineral oil have different freezing temperatures depending on the average molecular weight? Maybe the WISDOM technique used a different type of mineral oil? I suggest the authors add additional information on the conditions used in WISDOM and the conditions used in the current experiments.

**Molecular weight certainly plays a role in the freezing temperature of compounds. The difficulty with mineral oil is it is a mixture of dozens or hundreds of compounds derived from petroleum refining. There is no defined structure for mineral oils. Reicher et al. (2018) used mineral oil from Sigma Aldrich. Sigma has 11 different products called "mineral oil", all of which have the same CAS number despite their different physical and chemical properties. Reicher et al. almost certainly used a different mineral oil than we had, since we used mineral oil from VWR. We have included in this manuscript as much detail as was provided by Reicher et al. (2018) and believe we have adequately described the freezing of mineral oil found in this study. The relevant text has been modified as follows:**

**"However, the WISDOM microfluidic DFT device uses mineral oil for droplet creation and storage (Reicher et al., 2018). The device has successfully measured homogeneous ice nucleation down to -36 °C. Perhaps the surfactant (Span80, 2 wt%) used to stabilize the immersed droplets prevents freezing of the mineral oil. There are also a wide range of different mineral oils available from common chemicals suppliers and the specific type of oil used in WISDOM is not known. Alternatively, the optical fogging may not be visible when such a small volume of oil is above the droplets, as is the case for microfluidic devices. Despite the promising results from the WISDOM method, we are wary to suggest that any other groups attempt the use of mineral oil for droplet freezing measurements before further investigating how the oil's freezing may impact water droplet freezing. For all oil-immersion experiments mentioned in the following sections, squalene oil was used as the oil matrix, following the method of Wright and Petters (2013). Previously, we have shed light on squalene oil reducing the observed ice nucleation activity of Snomax bacterial particles and concluded this was due to hydrophobic partitioning of large protein aggregates (Polen et al., 2016). This was only observed in droplet refreeze experiments of Snomax, and we do not observe this effect on any other particle sample type we have tested. Squalene oil remains our recommended immersion oil for most droplet freezing experiments."**

Page 9, Line 299. This sentence suggests that the water was filtered for many weeks. I assume that this is not correct.

**Yes, this was unclearly worded. One filter was reused multiple times to filter several samples of HPLC water over the course of weeks. The phrasing has been changed as follows:**

**"The biggest deviation came from runs of HPLC water that was filtered multiple times over many weeks using the same Anotop filter."**

Page 10, Lines 317-319 and Page 19, Lines 581-582. The issues with the MilliQ produced water were blamed on the particle membrane filter. How do the authors know that the particle membrane filter was the source of the problem? I would have guessed any issue associated with the particle membrane filter would be rectified with the 0.02 micrometer post-filter.

**That is correct, with the addition of the Anotop 0.02 micrometer post filtering we should see similar results. However, we only saw an improvement once the MilliQ system particle filter was replaced. At this time we decided to switch to bottled HPLC water to not have to deal with any uncertainty and variability associated with the MilliQ system. We do not know why Anotop filtration did not prevent this contamination. We have clarified this in the text as follows:**

**"The MilliQ-produced water can result in very inconsistent results for pure water droplet controls if the particle membrane filter is not changed on a regular basis. These**

**contaminants were apparently not removed by filtering the poor-quality MilliQ water with a 20 nm pore Anotop filter, for reasons unknown to us."**

On Page 14, it was not clear how the Vaseline surfaces were made. Line 431 suggests Vaseline was spread on the sample dish. What was the sample dish made of? On page 19, Line 563, it sounds like the Vaseline was spread on the hydrophobic glass slides? Please clarify.

**The Vaseline was spread on the aluminum sample dish directly. The later statement on page 19 was meant to compare the Vaseline to hydrophobic coverslips as separate tests, not to imply that coverslips were used underneath the Vaseline. The phrasing of Line 561 has been changed to: "compared to hydrophobic cover slips".**

The format of the references needs to be improved in several cases.

**Thank you for pointing this out. The citations for Lohmann and Fleichter (2005), Wheeler et al. (2014), Wilson et al. (2015), and Tobo (2016) references have been corrected.**

---

## Author Comment (AC3) · 27 Aug 2018

Response to Reviewer #2

In this manuscript Polen et al. describe a study of non-homogeneous ice nucleation in 'pure' water samples and a variety of substrates. They then go on to make a sequence of suggestions for groups using droplet freezing assays. This is a highly valuable manuscript and it will help both newcomers to the field and established groups improve their droplet freezing techniques. I agree with the authors statement that individual groups have a great deal of knowledge on this subject, which isn't necessarily made publically available. The comments I have made below are made in this spirit. I support the manuscript's publication and list a few comments which I hope the authors will use to further strengthen their manuscript:

**We thank the referee Ben Murray for his positive response to our manuscript and for his detailed comments that we have used to further improve the quality and clarity of our manuscript. Below we respond to each comment in turn and describe any revisions made accordingly.**

Handling blank experiments vs. 'pure' water control experiments. In the list of recommendations, I encourage the authors to include a recommendation that experimentalists conduct full handling blank experiments in addition to 'pure' water control experiments. By handling blank experiment I mean putting the water used in the experiment through the full process that the water containing a sample had been through. In Vergara-Temprado et al. (1) we demonstrated that this was critical. The 'pure' water control experiments were lower than the handling blanks (see Figure 2). The experiments where droplets were loaded with black carbon froze at a similar temperature to the handling blanks, but were above the pure water control. We consequently reported limiting values. In Fig 12, is the control experiment a handling blank?

**This is a very useful and important point. We agree that the use of handling blanks (also referred to as a "method blank" or "field blank") is important and have added this to our list of recommendations under number 7. We have also added a brief discussion of the importance of method blanks as recently reported by Vergara-Temprado et al. In Figure 12 we are using a control, not a handling or method blank, but method blanks will be tested and reported for future experiments of this nature.**

Abstract: The word 'plagued' is perhaps a bit strong. It is a limitation.

**We agree, the wording has been changed to: "These droplet freezing experiments are often limited by contamination"**

The authors refer to homogeneous nucleation at -38 C and the 'homogeneous nucleation limit'. This creates the impression that homogeneous nucleation has a well-defined limit where it occurs. It is volume and time dependent and this matters. For example, a 1 um droplet cooled at 1 K min-1 will have a freezing probability of 0.5 at -33.5C (according to Koop and Murray (2)). Furthermore Herbert et al.(3) found that enough cloud droplets started to freeze in a cloud to start

to affect cloud properties at around -33 C or so (depending on the homogeneous parameterisation), even though 50% of droplets would only freeze homogeneously at around -38 C. I would like to see the introductory sentences adjusted to be less definitive about the when homogeneous nucleation becomes important. Also, at ln 51, define the 'homogeneous nucleation limit' as, for example, the T at which 50% of 10 um droplets are expected to freeze on cooling at 1 K min-1.

**We agree and appreciate the suggestion. We have discussed the homogeneous freezing temperature in an artificially simplistic binary manner. We have changed the term "limit" to "regime" and cite Koop and Murray (2016) for a detailed discussion of the homogeneous freezing rate and thus freezing probability versus temperature and the relevant equations. This seems more appropriate for the purpose of our manuscript. This has been changed throughout the manuscript, such as in the Introduction as follows:**

**"Water contamination or substrate interferences can also induce freezing well above the homogeneous freezing temperature regime that ensues in the temperature range of -35 to -40 °C (Koop and Murray, 2016), restricting the heterogeneous temperature regime accessible by DFTs."**

Ln 34-35. Vergara-Temprado et al. (4) could be cited here, this paper clearly shows a sensitivity of clouds to INP.

**This citation has been added.**

Ln78. Replace 'steals' is not the best choice of words.

**Phrasing has been changed to: "The WBF process occurs when one droplet freezes and takes up water vapor at the expense of unfrozen droplets…"**

Ln93-97: Tarn et al. (5) also used microfluidic technology to study heterogeneous freezing. One of the objectives was to see if the oil and surfactant influenced nucleation. They measured ice nucleation (ns) for a range of materials and compared to literature. The results suggest that these technologies can be used to make droplets and study heterogeneous nucleation.

**Thank you for suggesting this very recent relevant publication. This citation has been added and we briefly discuss their relevant findings regarding microfluidic droplet generation for ice nucleation measurements as follows:**

**"Some recent microfluidic ice nucleation techniques use fluorinated oils and/or large concentrations of surfactant to stabilize the emulsified droplets (Reicher et al., 2018; Stan et al., 2009; Tarn et al., 2018). Their measured homogeneous freezing temperatures are typically within the expected range (-35 to -37 °C), but the surfactant may have**

**unrecognized influences on heterogeneous freezing processes since freezing is enhanced via contact between the immersed particle and droplet interface** (Durant and Shaw, 2005; Futuka, 1975; Gurganus et al., 2014; Tabazadeh et al., 2002)**. However, Tarn et al. (2018) concluded that surfactants seemed to have little effect on the observed heterogeneous freezing temperatures for a number of particle types examined using droplets prepared by microfluidics with added surfactant."**

Ln 214. 'correct for', replace with 'account for'. This isn't a correction.

**Phrasing has been changed.**

Fig 2 and discussion. These images are similar to those shown in Fig 4 or Whale et al. (6), mention that the new findings are consistent with what was previously found. Also, note that it is relatively easy to see if this is a problem.
Also, just for this discussion: we have found that this becomes a more significant problem when doing experiments in a humid environment. We solved this problem by improving the design of our ul-NIPI chamber to make it more air tight.

**Yes we also see that this issue is exacerbated in the summer months when humidity is much higher. We have discussed this in the manuscript so that other groups are aware of this issue and can design their systems to reduce it, as mentioned above. The citation has been added and we discuss the similarity of their results as follows:**

**"Frost growth similar to this has been reported previously by Whale et al.** (Whale et al., 2015) **in their cold stage system. This suggests that our system is not air tight enough to perform this type of experiment when ambient humidity levels are elevated such as during summer."**

'Particle sedimentation out of the droplet'. I don't think this is very likely. They may sediment to the bottom of a droplet, but are unlikely to sediment across an interface. Emersic et al. provide no evidence that particles can sediment out of a droplet.

**This statement has been changed as not to suggest that particles are likely to cross the surface layer of the droplet. Until direct measurements of this phenomenon are reported we feel that we cannot be certain that no sedimentation out of the water droplet occurs, especially since there is a layer of oil beneath the water droplets in our microfluidic device. We have changed this as follows:**

**"…and any time-dependent effects such as particle sedimentation or aggregation."**

P 9-10. We have also compared HPLC water and compared to water from MiliQ systems. We have found that the quality of HPLC water is also variable. I would recommend that whatever the source of water, the experimenter should demonstrate its quality and do the experiments to test the quality at sufficient intervals. I also agree that water from MiliQ machines can be highly variable, but in our experiments if the machine is well-maintained then the quality is systematically high, although we too have had periods when the quality was much lower.

**This is an important perspective and a major reason why we believe more open discussion should be had within the ice nucleation community on the sources of water and effective strategies to reduce water contamination and related effects. We have added this insight to point 4 in the list as follows:**

**"Interestingly, we have also heard that other research groups found bottled water is not as consistent as their MilliQ-produced water. This demonstrates the inconsistencies and variabilities that are common between research groups and suppliers, further emphasizing the importance of routinely assessing and reporting the water background freezing spectrum that each group and method observes. We suggest that no matter what source of water is used that researchers regularly test it and report their findings in all publications when possible."**

Fraction frozen plots throughout. I think it would be valuable to show the theoretical homogeneous nucleation curve. I would suggest using Koop and Murray (2) since in this paper the authors attempted to constrain classical theory in a physically plausible way, which gives more confidence in the values at higher temperatures (relevant for ul sized droplets) where there is very little or no data.

**We have added homogeneous nucleation curves to each of the droplet freezing spectra for the relevant droplet volumes using equations A9a and A9b from** (Koop and Murray, 2016) **to calculate J$_{homogeneous}$ and a freezing probability derived similarly to that calculated using Equation 9 from Beydoun et al. (2016):**

$$P_f = 1 - exp(-\left(\frac{V}{\dot{T}}\right) * \int_{T_0}^{T_{hom}} JdT$$

**where $\dot{T}$ is the cooling rate, $V$ is the droplet volume, and $T_{hom}$ and $T_0$ are the bounds that encompass the full freezing probability used here of 250 K to 220 K.**

**We describe this in Section 3:**

**"We include the theoretical homogeneous freezing spectrum for our droplet sizes in all our droplet freezing temperature spectra below. This was produced using the parameterization of Koop and Murray (2016) to calculate the freezing rate, $J(T)$, and Eqn. 9 from Beydoun et al. (2016) to determine the frozen fraction, $P_f$, using $J(T)$."**

Substrate dependent nucleation: Mention the result of Price et al. (7) (Figure 4) where it was shown that a Teflon substrate produced lower freezing temperatures when compared to a salinized glass surface.

**This is nice additional information that further supports our arguments regarding the importance of substrate effects on initiating freezing at artificially warmer temperatures. We have added a brief discussion of this paper's results as follows:**

**"Price et al.** (Price et al., 2018) **reported observing lower freezing temperatures when droplets were placed on a Teflon substrate compared to on a standard silanized hydrophobic glass surface. This provides further support for the important role that substrate choice can have on the freezing temperature spectrum observed in droplet freezing techniques."**

Ln 713: On the topic of K-feldspar being sensitive to water. We explored this in Harrison et al. (8) and showed that BCS 376 (a standard feldspar which is available for anyone to buy) only degraded by _1 C over 16 months in water. Other feldspars, in particular those exhibiting hyperactivity, are more sensitive to time in water.

**Thank you for the suggestion. We have added the citation and brief discussion of these findings, as follows:**

**"The very ice-active K-feldspar minerals are especially subject to degradation in water due to surface ion etching, particularly for those displaying hyperactive ice-activity (Banfield and Eggleton, 1990; Holdren and Berner, 1979; Kumar et al., 2018; Peckhaus et al., 2016). Harrison et al.** (Harrison et al., 2016) **found that a particular and common type of feldspar that does not display hyperactivity, BCS 376, was able to maintain its IN activity over many months in water. Engineered nanoparticles from inert metal oxides with reproducible particles sizes, surface properties, and pore sizes may be the most reliable type of INP standard, though this has not yet been evaluated and may be restricted to a narrow freezing temperature range (Alstadt et al., 2017; Archuleta et al., 2005; Findenegg et al., 2008; Marcolli et al., 2016)."**

Also, I think Vali's comment on this paper is really valuable, and it would be useful to record this approach in the literature (either in Polen et al. or Vali could consider a short note in AMT setting this out formally?).

**We absolutely agree that Gabor Vali's comment provides important insight and should be shared more widely with the ice nucleation community. We have added a discussion of his suggestions regarding correcting the c_INP(T) or K(T) spectrum using 'pure' water control data through an iterative (k(T)) versus cumulative (K(T)) approach, and cited his comment on our manuscript (doi: 10.5194/amt-2018-134-SC1). We have also had extensive discussions with Gabor Vali regarding the k(T) approach. Through this he realized there**

are some important but not obvious details involved in properly using k(T), such as the width of the temperature step $\Delta T$ used. He plans to write a tutorial on this, using some of our droplet freezing data to illustrate the correct application of the analysis. We will also demonstrate the use of k(T) in a forthcoming paper that describes the design and evaluates the performance of our new microfluidic droplet freezing approach.

The following discussion was added to the manuscript:

"Alternatively, retrieval of the differential nucleus concentration, referred to as $k(\theta)$ in Vali (1971), is also recommended to assess the INP concentration in the sample versus that caused by background freezing. This approach can be used as a means of quantitatively attributing the INP signal to the sample versus the background for each droplet over the entire freezing spectrum. The differential nucleus concentration can be calculated using:

$$k(T) = -1/(V_d * \Delta T) * \ln[1 - \Delta N/N(T)] \quad (2)$$

where k(T) is the differential ice nucleus concentration, $V_d$ is the droplet volume, $\Delta T$ is a temperature step that must be prescribed in the analysis, $\Delta N$ is the number of droplets that froze in that $\Delta T$ temperature step, and N(T) is the total number of unfrozen droplets at T. An important aspect is that $\Delta T$ is **not** the temperature step of the actual measurements, such as from the frequency at which images are acquired. To produce meaningful k(T) spectra the $\Delta T$ should be large enough such than more than one droplet typically freezes in a given temperature step. In our initial k(T) analysis we found a $\Delta T$ interval of 0.05 or 0.1 °C to work well for our experimental conditions. $\Delta T$ should be varied until a reasonable representation of the droplet freezing spectrum is produced that displays the important features of the spectrum and allows the sample to be distinguished from the background freezing of a control. Realizing that this is an important and nuanced detail, Gabor Vali is planning to produce a tutorial explaining the use of k(T) and selection of $\Delta T$, using some of our data to illustrate this method. Referring back to Eq. (2), as an example, given an array of 100 droplets and a specified $\Delta T$ of 0.1 °C intervals, if the first 2 droplets freeze within one measurement interval, $\Delta T = 0.1$ °C, $\Delta N = 2$, and N(T) = 98. Using this metric, each freezing event in the interval $\Delta T$ is the result of at least one active INP, but given a small $\Delta T$ and a large N the interval can be approximately attributed to a single active INP.

Inherent to all droplet freezing methods is the assumption that the freezing of any droplet at a given temperature interval is caused by the combination of INPs present from the sample plus any background freezing due to impurities and substrate artifacts. The differential ice nucleus method, k(T), provides a quantitative assessment of the sample versus the background INP concentration at each temperature interval. k(T) is an alternative approach to the more commonly used method of just subtracting the cumulative K(T) or $c_{INP}$ background spectrum from the cumulative sample spectrum. This k(T) analysis method is discussed in detail by Gabor Vali in the comment (doi: 10.5194/amt-2018-134-SC1) he provided on the discussion version of this manuscript (https://www.atmos-meas-tech-discuss.net/amt-2018-134/amt-2018-134-SC1-supplement.pdf), based on the framework originally laid out in Vali (1971)."

**References**

[revised manuscript text omitted]

---

## Author Comment (AC4) · 27 Aug 2018

We thank Dr. Vali for his extensive and highly insightful comments. The k(T) derivative method of analyzing droplet freezing spectra is certainly a valuable approach that is not typically used by the ice nucleation community. Making the community aware of this alternative analysis approach first introduced by Vali (1971) is a worthwhile endeavor and thus we have included a discussion of this in the revised manuscript. We note there are two small typos in the original comment provided by Gabor Vali that we would like to correct here for clarity: Eq. 1 that is referred to from Vali (1971) is actually Eq. 11, and $\Delta T$ was accidentally used instead of $\Delta N$ in the natural logarithm term. Here is the

original Eq. 11 from Vali (1971), using T instead of theta for temperature:

$k(T) = -1/(V_d*\Delta T)*\ln[1-\Delta N/N(T)]$

where T is temperature, V_d is the droplet volume, $\Delta N$ is the number of drops that froze during the temperature interval $\Delta T$, and N(T) is the number of drops still unfrozen at temperature T.

We have engaged in a series of discussions with Gabor Vali regarding the use of k(T) to analyze our droplet freezing spectra. Through this discussion Gabor realized there are some important but not obvious details involved in properly using k(T), such as the width of the temperature step $\Delta T$ (or $\Delta$theta) used. $\Delta T$ should be large enough such that more than one droplet freezing event occurs during a $\Delta T$ interval, but not so large such that important features of the k(T) spectrum are not observable. In our initial k(T) analysis we found a $\Delta T$ of 0.05 or 0.1 C to be an appropriate choice. Gabor plans to write a tutorial fully describing this analysis, using some of our droplet freezing data to illustrate the correct application of the k(T) method. We will also demonstrate the use of k(T) in a forthcoming paper that describes the design and evaluates the performance of our new microfluidic droplet freezing approach. To add k(T) analysis to this current manuscript would require rather significant additions to fully explain and illustrate this rather nuanced analysis. Instead we have added a detailed discussion of the k(T) analysis method with a link to Gabor's original comment, as follows:

"Alternatively, retrieval of the differential nucleus concentration, referred to as $k(\theta)$ in Vali (1971), is also recommended to assess the INP concentration in the sample versus that caused by background freezing. This approach can be used as a means of quantitatively attributing the INP signal to the sample versus the background for each droplet over the entire freezing spectrum. The differential nucleus concentration can be calculated using:

$k(T) = -1/(V_d*\Delta T)*\ln[1-\Delta N/N(T)]$ (2)

where k(T) is the differential ice nucleus concentration, $V\_d$ is the droplet volume, $\Delta T$ is a temperature step that must be prescribed in the analysis, $\Delta N$ is the number of droplets that froze in that $\Delta T$ temperature step, and N(T) is the total number of unfrozen droplets at T. An important aspect is that $\Delta T$ is not the temperature step of the actual measurements, such as from the frequency at which images are acquired. To produce meaningful k(T) spectra the $\Delta T$ should be large enough such than more than one droplet typically freezes in a given temperature step. In our initial k(T) analysis we found a $\Delta T$ interval of 0.05 or 0.1 °C to work well for our experimental conditions. $\Delta T$ should be varied until a reasonable representation of the droplet freezing spectrum is produced that displays the important features of the spectrum and allows the sample to be distinguished from the background freezing of a control. Realizing that this is an important and nuanced detail, Gabor Vali is planning to produce a tutorial explaining the use of k(T) and selection of $\Delta T$, using some of our data to illustrate this method. Referring back to Eq. (2), as an example, given an array of 100 droplets and a specified $\Delta T$ of 0.1 °C intervals, if the first 2 droplets freeze within one measurement interval, $\Delta T = 0.1$ °C, $\Delta N = 2$, and N(T) = 98. Using this metric, each freezing event in the interval $\Delta T$ is the result of at least one active INP, but given a small $\Delta T$ and a large N the interval can be approximately attributed to a single active INP.

Inherent to all droplet freezing methods is the assumption that the freezing of any droplet at a given temperature interval is caused by the combination of INPs present from the sample plus any background freezing due to impurities and substrate artifacts. The differential ice nucleus method, k(T), provides a quantitative assessment of the sample versus the background INP concentration at each temperature interval. k(T) is an alternative approach to the more commonly used method of just subtracting the cumulative K(T) or cINP background spectrum from the cumulative sample spectrum. This k(T) analysis method is discussed in detail by Gabor Vali in the comment (doi: 10.5194/amt-2018-134-SC1) he provided on the discussion version of this manuscript (https://www.atmos-meas-tech-discuss.net/amt-2018-134/amt-2018-134-SC1- supplement.pdf), based on the framework originally laid out in Vali (1971)."

Reference:

Vali, G.: Quantitative Evaluation of Experimental Results an the Heterogeneous Freezing Nucleation of Supercooled Liquids, J. Atmos. Sci., 28(3), 402–409, doi:doi.org/10.1175/1520-0469(1971)028<0402:QEOERA>2.0.CO;2, 1971.

––––––––––––––––––––––––

---

## Author Response (AR1)

[revised manuscript text omitted]

We have attempted droplet-in-air measurements within our own system but consistently had issues with frost halo formation upon reaching -20 °C using a standard cooling rate of 1 °C/min (Budke and Koop, 2015; Jung et al., 2012). A series of images in Figure 2 shows this frost growth, which resulted in freezing of nearly all pure water droplets by -20 °C on hydrophobic coverslips when oil wasn't used. Frost growth similar to this has been reported previously by Whale et al. (2015) in their cold stage system. This suggests that our system is not air tight enough to perform this type of experiment when ambient humidity levels are elevated such as during summer. Li et al. (2012) froze their samples between two glass cover slides which were sealed together with vacuum grease for the entire experiment. Our chamber must be opened between runs which causes water vapor to condense onto the sample dish and elsewhere within the sample chamber. In this experiment, we had dry nitrogen flushing the chamber similar to previous methods but frost growth still occurred, though at much lower temperatures than tests without the nitrogen flow. Figure 2 shows the progression of frost starting at the bottom of the cover slip and continuing to grow toward the top of the glass. We consistently found that freezing and frost growth initiated around -20 °C, and we were never able to approach homogeneous freezing, likely due to our slow but realistic 1 °C/min cooling rate.

[Figure]

Figure 2. Progression of frost halos in one pure water droplet freezing experiment without an oil matrix. Dark droplets are frozen. The black line highlights the frost growth (which is visible in the image but difficult to see) spreading from the bottom left toward the top of the image. Aside from the indicated frost growth, we can also see that other droplets induce freezing in neighboring droplets, such as the droplet on the far right in image 1 (red arrow) and the top right droplet in image 2 (yellow arrow). Subsequently induced droplets are indicated by similarly colored arrows.

Many droplet freezing measurements use an oil matrix to prevent frost halos, droplet evaporation, and external contamination (Broadley et al., 2012; Pummer et al., 2015; Wright et al., 2013; Zolles et al., 2015), which is why we chose to use squalene oil for our measurements. Oil also facilitates droplet refreeze experiments to evaluate the repeatability of the ice nucleation process, and any time-dependent effects such as particle sedimentation or aggregation (Emersic et al., 2015; Wright et al., 2013). In Polen et al. (2016), we proposed the use of mineral oil for biological samples, such as Snomax, to prevent changes in freezing behavior due to hydrophobic partitioning, which we suspected to be the case for refreezes performed in squalene oil ($C_{30}H_{50}$). However, in our attempts to use mineral oil (light) in pure water measurements the mineral oil froze around -30 °C. We consistently saw what we at first assumed to be fogging, but upon closer inspection we found that the mineral oil had frozen completely solid, precluding droplet freezing experiments. Though we never saw mention of the freezing point in the material safety data sheets provided for the mineral oils, this is a known issue in the use of mineral oil for liquid chilling in desktop computers. However, the WISDOM microfluidic DFT device uses mineral oil for droplet creation and storage (Reicher et al., 2018). The device has successfully measured homogeneous ice nucleation down to -36 °C. Perhaps the surfactant (Span80, 2 wt%) used to stabilize the immersed droplets prevents freezing of the mineral oil. There are also a wide range of different mineral oils available from common chemicals suppliers and the specific type of oil used in WISDOM is not known. Alternatively, the optical fogging may not be visible when such a small volume of oil is above the droplets, as is the case for microfluidic devices. Despite the promising results from the WISDOM method, we are wary to suggest that any other groups attempt the use of mineral oil for droplet freezing measurements before further investigation into how the oil's freezing may impact water droplet freezing. For all oil-immersion experiments mentioned in the following sections, squalene oil was used as the oil matrix, following the method of Wright and Petters (2013). Previously, we have shed light on squalene oil reducing the observed ice nucleation activity of Snomax bacterial particles and concluded this was due to hydrophobic partitioning of large protein aggregates (Polen et al., 2016). This was only observed in droplet refreeze experiments of Snomax, and we do not observe this effect on any other particle sample type we have tested. Squalene oil remains our recommended immersion oil for most droplet freezing experiments.

**4.2 Water sources and purification**

Many in the ice nucleation community use MilliQ water or similar commercial systems to purify their laboratory's in-house water (Inada et al., 2014; Pummer et al., 2015; Rigg et al., 2013; Tobo, 2016; Umo et al., 2015; Wright and Petters, 2013). Some groups have used bottled HPLC grade or other similar water for their DFT (Fornea et al., 2009; Wright and Petters, 2013). Still others use alternative methods, such as condensation, to create droplets (Campbell et al., 2015; Li et al., 2012; Mason et al., 2015). We compared water produced by our in-house MilliQ system with bottled HPLC-grade water from Sigma Aldrich (Figure 3). Both water types were also filtered using 0.02 µm pore size Anotop filters before droplet generation. In general, the droplet freezing spectra obtained from the two types of water are very similar to one another. With ~1000 droplets for each water type, we find little difference in the apparent INP concentration as well. The biggest deviation came from runs of HPLC water that was filtered multiple times over many weeks using the same Anotop filter, which shows an increase in ice nuclei around -25 °C, though this is not outside the standard deviation of our other samples. This result indicates that either purchased HPLC or produced MilliQ water could be useable for droplet freezing experiments. As MilliQ water systems use a series of filter cartridges and a membrane filter to remove dissolved contaminants, particles, and ions from the supplied water, the quality of the produced water achieved will depend on the quality of the original water supply source. The "house" water supply is beyond the control of most research groups. Along with other issues we have experienced using MilliQ water that we discuss below, high-quality bottled water with additional filtration may be a better and more reliable water source for ice nucleation studies.

[revised manuscript text omitted]

We have also observed some batches of purchased coverslips to induce freezing as warm as -18 °C, and with much greater variability in the freezing spectra. Thus, it is important to evaluate each batch of coverslips to test for these potential issues. Ideally pure water control droplets will be placed along with droplets containing the particle sample of interest on the *same* coverslip to directly evaluate the background freezing spectrum on that specific cover slip. This is especially important when working with particle systems of weak ice-activity that freeze close to the background water temperature range.

4.3.2 Silicon wafers

A few groups have utilized silicon wafers for droplet freezing experiments (Li et al., 2012; Peckhaus et al., 2016). Peckhaus et al. (2016) used droplets of 107 μm in diameter and found 90% of droplets froze below -35 °C. All droplets reported by Li et al. (2012) froze below -37.5 °C for 10-70 μm in diameter. Additionally, Li et al. performed detailed assessment of hydrophobic and hydrophilic silicon wafers used in pure water ice nucleation experiments. They found that both types of wafer produced nearly homogeneous freezing for pure water droplets.

We investigated ice nucleation on silicon wafer chips typically used for SEM analysis. Several silicon chips were placed in the sample dish with squalene oil, and 0.1 uL (~600 μm) HPLC droplets were deposited on them. Due to the small size (5x7 mm) of the chips, the number of droplets on each wafer chip was very low (~10), and thus we combined all the data from twelve chips as though it were a single surface containing 120 droplets (Fig. 6). We find similar freezing activity to the hydrophobic cover slips with onset freezing beginning around -21 °C, reaching 50% around -26 °C, and finishing at -35 °C. The apparent INP concentration for the silicon wafer also falls close to the cover slip data (Fig. 6). We are using much larger droplets (~6-60x diameter) than the groups who have used silicon substrates previously, so we do see higher freezing temperatures as expected. However, due to the similar behavior and apparent INP concentration we observe using the glass cover slips and the silicon wafer, we cannot conclude that silicon provides a more ideal surface for INP studies than silanized hydrophobic glass. The superior performance reported by other groups using silicon wafers may be due to higher purity water than we have access to, or other method details that make a direct comparison challenging.

[Figure]

**Figure 6.** Comparison of freezing on silicon wafer chips (green) against hydrophobic cover slips (blue), following Figure 5. The freezing temperature spectrum is on the left, and the retrieved $c_{IN}$ is on the right. Both datasets use 0.1 µL droplets. The data from all replicate arrays using silicon (green) are combined into one series and thus no error bars can be determined. The parentheses next to each legend entry contains the number of arrays of droplets (A) and the total number of droplets across all arrays (N). The gray dashed line indicates the theoretical homogeneous freezing curve of 0.1 µL droplets, using the parameterization of Koop and Murray (2016).

**4.3.3 Vaseline®**

First utilized by Tobo (2016) for the Cryogenic Refrigerator Applied to Freezing Test (CRAFT) droplet freezing instrument, Vaseline® petroleum jelly can be spread onto a clean surface to create a makeshift hydrophobic substrate. The results from Tobo (2016) indicate great promise in this substrate for DFT as the large, 5 µL droplets froze with $N_{50}$ = -33 °C, approaching the temperature predicted by CNT for homogeneous freezing. We examined large (1.0 µL) droplets on Vaseline® spread onto our aluminum sample dish in air, similar to Tobo (2016), as well as smaller droplets (0.1 µL) on Vaseline®, and within a squalene oil matrix. The results are shown in Figure 7. For tests without the oil matrix, we found quite warm onset freezing temperatures while only a few droplets approached the homogeneous regime. We found similar trends whether we used MilliQ water or filtered HPLC water. However, once we utilized smaller droplets in an oil matrix, the early onset freezing vanished and we observed good background freezing curves with lower onset and $N_{50}$ temperatures. We hypothesize that our inability to reproduce pure water freezing near the homogeneous regime using a Vaseline® coated substrate as in Tobo (2016) is due to the difference in cleanliness between laboratory environments as well as differences in applying the Vaseline® layer. The oil matrix does eliminate much of the early, high temperature freezing that is likely caused by contamination or an unevenly coated surface. This suggests the use of a laminar flow hood or glove box may be necessary to achieve such low background freezing temperatures without oil when the droplets are exposed to air. Tobo prepared their droplet arrays inside a glove box within a clean room environment, and such clean conditions are not readily available to many research groups. Uniform application of Vaseline® requires precision and a specialized spatula to get around the lipped design, and non-uniform application will increase the risk of surface-induced freezing by any exposed underlying substrate. Interestingly, we note that one benefit to Vaseline® is we did not observe evidence of WBF effects on neighboring droplets
when in air, which makes it favorable for droplets-in-air experiments if interferences can be
reduced. Creation of a surface specifically designed for Vaseline® application is an important
consideration if this promising technique is to be utilized more widely.

[revised manuscript text omitted]

PDMS (Fig. 9). The pure water freezing spectra are again similar to our silanized cover slip results, as we have seen for most of the other substrates tested. Each of the PDMS tests was within the standard deviation of the CS data, suggesting that the PDMS surface does not provide any inherent benefit over hydrophobic silanized glass. On the other hand, PDMS is quite cheap and easy to manipulate if you have the resources to do so, which makes it a quite useful substrate for IN studies. The hydrophobic nature of the polymer can make it prone to contamination however, and PDMS is often used as a sorbent in environmental contaminant sampling (Choi et al., 2011; Thomas et al., 2014). One other potential downside to PDMS for DFTs is its poor heat transfer properties. The thickness of the PDMS layer must be consistent for each experiment or the temperature calibration will be inaccurate.

[Figure]

**Figure 9.** Measurements of HPLC pure water droplet freezing on PDMS are shown in red and green, following Figure 5. The data from small droplets on a silanized coverslip are displayed for comparison in blue (Fig. 5, blue). The PDMS data was obtained using treated (red) and untreated (green) PDMS polymer with small droplets. The parentheses next to each legend entry contains the number of arrays of droplets (A) and the total number of droplets across all arrays (N). Error bars on green data show standard deviation from replicate arrays, while the red data are combined into one series as explained in section 2. The gray dashed line indicates the theoretical homogeneous freezing curve of 0.1 µL droplets, using the parameterization of Koop and Murray (2016).

We have recently developed a new "store-and-create" microfluidic device that shows great promise in eliminating the interferences from surface interactions as seen in our and other groups' DFTs (Bithi and Vanapalli, 2010; Boukellal et al., 2009; Sun et al., 2011). This device will be fully described in a forthcoming manuscript. The PDMS device holds up to 600 droplets of ~6 nL volume encased in squalene oil. Each droplet is stored in an isolated microwell, completely engulfed by oil. Initial results for pure water droplet freezing are shown in Figure 10 and compared with hydrophobic silanized cover slips. We find a $N_{50}$ around -34 °C with less than 10% of droplets freezing above -32 °C. Interestingly, we see that the apparent INP concentration continues the same trend as the 0.1 uL droplets on a hydrophobic cover slip. This is likely because the droplets lack contact with any solid surface inside the microfluidic device and the contaminants causing this non-homogeneous freezing are related to water or oil contaminants.

[Figure]

**Figure 10.** Comparison of pure water droplet freezing in our new microfluidic chip (red) versus using a silanized cover slip (CS) (blue), following Figure 5. Droplets in the microfluidic chip are 6 nL in volume and droplets on the CS are 0.1 µL. Error bars show variability of droplet freezing between different replicate arrays. The parentheses next to each legend entry contains the number of arrays of droplets (A) and the total number of droplets across all arrays (N). The gray and brown dashed lines indicate the theoretical homogeneous freezing curves of 0.1 µL and 6 nL droplets, respectively, using the parameterization of Koop and Murray (2016).

[revised manuscript text omitted]

Alternatively, retrieval of the differential nucleus concentration, referred to as $k(\theta)$ in Vali (1971), is also recommended to assess the INP concentration in the sample versus that caused by background freezing. This approach can be used as a means of quantitatively attributing the INP signal to the sample versus the background for each droplet over the entire freezing spectrum. The differential nucleus concentration can be calculated using:

$$k(T) = -1/(V_d * \Delta T) * \ln[1 - \Delta N / N(T)] \qquad (2)$$

where $k(T)$ is the differential ice nucleus concentration, $V_d$ is the droplet volume, $\Delta T$ is a temperature step that must be prescribed in the analysis, $\Delta N$ is the number of droplets that froze in that $\Delta T$ temperature step, and $N(T)$ is the total number of unfrozen droplets at $T$. An important aspect is that $\Delta T$ is not the temperature step of the actual measurements, such as from the frequency at which images are acquired. To produce meaningful $k(T)$ spectra the $\Delta T$ should be large enough such than more than one droplet typically freezes in a given temperature step. In our initial $k(T)$ analysis we found a $\Delta T$ interval of 0.05 or 0.1 °C to work well for our experimental conditions. $\Delta T$ should be varied until a reasonable representation of the droplet freezing spectrum is produced that displays the important features of the spectrum and allows the sample to be distinguished from the background freezing of a control. Realizing that this is an important and nuanced detail, Gabor Vali is planning to produce a tutorial explaining the use of $k(T)$ and selection of $\Delta T$ using some of our data to illustrate this method. Referring back to Eq. (2), as an example, given an array of 100 droplets and a specified $\Delta T$ of 0.1 °C intervals, if the first 2 droplets freeze within one measurement interval, $\Delta T = 0.1$ °C, $\Delta N = 2$, and $N(T) = 98$. Using this metric, each freezing event in the interval $\Delta T$ is the result of at least one active INP, but given a small $\Delta T$ and a large $N$ the interval can be approximately attributed to a single active INP.

Inherent to all droplet freezing methods is the assumption that the freezing of any droplet at a given temperature interval is caused by the combination of INPs present from the sample plus any background freezing due to impurities and substrate artifacts. The differential ice nucleus method, $k(T)$, provides a quantitative assessment of the sample versus the background INP concentration at each temperature interval. $k(T)$ is an alternative approach to the more commonly used method of just subtracting the cumulative $K(T)$ or $c_{INP}$ background spectrum from the cumulative sample spectrum. This $k(T)$ analysis method is discussed in detail by Gabor Vali in the comment (doi: 10.5194/amt-2018-134-SC1) he provided on the discussion version of this manuscript (https://www.atmos-meas-tech-discuss.net/amt-2018-134/amt-2018-134-SC1-supplement.pdf), based on the framework originally laid out in Vali (1971).

[revised manuscript text omitted]

*Acknowledgements.* We thank Tom Hill at Colorado State University for valuable suggestions and providing the Anotop filters for our testing. Hassan Beydoun and Leif Jahn provided comments on a draft of the manuscript. Comments provided by Gabor Vali, Benjamin Murray, and an anonymous referee during review significantly improved this 
[revised manuscript text omitted]

Petters, M. D. and Wright, T. P.: Revisiting ice nucleation from precipitation samples, , 1–9, doi:10.1002/2015GL065733.Received, 2015.

Polen, M., Lawlis, E. and Sullivan, R. C.: The unstable ice nucleation properties of Snomax® bacterial particles, J. Geophys. Res. Atmos., 121(19), 11,666-11,678,
doi:10.1002/2016JD025251, 2016.

Polen, M., Brubaker, T., Somers, J. and Sullivan, R. C.: Cleaning up our water: reducing interferences from non-homogeneous freezing of "pure" water in droplet freezing assays of ice nucleating particles, Atmos. Meas. Tech. Discuss., 1–31, doi:10.5194/amt-2018-134, 2018.

Price, H. C., Baustian, K. J., McQuaid, J. B., Blyth, A., Bower, K. N., Choularton, T., Cotton, R.

J., Cui, Z., Field, P. R., Gallagher, M., Hawker, R., Merrington, A., Miltenberger, A., Neely III, R. R., Parker, S. T., Rosenberg, P. D., Taylor, J. W., Trembath, J., Vergara-Temprado, J., Whale, T. F., Wilson, T. W., Young, G. and Murray, B. J.: Atmospheric Ice-Nucleating Particles in the Dusty Tropical Atlantic, J. Geophys. Res. Atmos., 123(4), 2175–2193, doi:10.1002/2017JD027560, 2018.

[revised manuscript text omitted]

Vergara-Temprado, J., Holden, M. A., Orton, T. R., O'Sullivan, D., Umo, N. S., Browse, J., Reddington, C., Baeza-Romero, M. T., Jones, J. M., Lea-Langton, A., Williams, A., Carslaw, K. S. and Murray, B. J.: Is Black Carbon an Unimportant Ice-Nucleating Particle in Mixed-Phase Clouds?, J. Geophys. Res. Atmos., 123(8), 4273–4283, doi:10.1002/2017JD027831, 2018a.

Vergara-Temprado, J., Miltenberger, A. K., Furtado, K., Grosvenor, D. P., Shipway, B. J., Hill,
A. A., Wilkinson, J. M., Field, P. R., Murray, B. J. and Carslaw, K. S.: Strong control of Southern Ocean cloud reflectivity by ice-nucleating particles, Proc. Natl. Acad. Sci., 115(11), 2687–2692, doi:10.1073/pnas.1721627115, 2018b.

Wang, B., Knopf, D. A., China, S., Arey, B. W., Harder, T. H., Gilles, M. K. and Laskin, A.: Direct observation of ice nucleation events on individual atmospheric particles, Phys. Chem.
Chem. Phys., 18(43), 29721–29731, doi:10.1039/C6CP05253C, 2016.

Wex, H., Augustin-Bauditz, S., Boose, Y., Budke, C., Curtius, J., Diehl, K., Dreyer, A., Frank, F., Hartmann, S., Hiranuma, N., Jantsch, E., Kanji, Z. a., Kiselev, A., Koop, T., Möhler, O., Niedermeier, D., Nillius, B., Rösch, M., Rose, D., Schmidt, C., Steinke, I. and Stratmann, F.: Intercomparing different devices for the investigation of ice nucleating particles using Snomax®
as test substance, Atmos. Chem. Phys., 15(3), 1463–1485, doi:10.5194/acp-15-1463-2015, 2015.

Whale, T. F., Murray, B. J., O'Sullivan, D., Wilson, T. W., Umo, N. S., Baustian, K. J., Atkinson, J. D., Workneh, D. A. and Morris, G. J.: A technique for quantifying heterogeneous ice nucleation in microlitre supercooled water droplets, Atmos. Meas. Tech., 8(6), 2437–2447, doi:10.5194/amt-8-2437-2015, 2015.

Wheeler, M. J., Mason, R. H., Steunenberg, K., Wagstaff, M., Chou, C. and Bertram, A. K.: Immersion Freezing of Supermicron Mineral Dust Particles: Freezing Results, Testing Different Schemes for Describing Ice Nucleation, and Ice Nucleation Active Site Densities, J. Phys. Chem. A, 119(19), 4358–4372, doi:10.1021/jp507875q, 2015.

Wilson, T. W., Ladino, L. A., Alpert, P. A., Breckels, M. N., Brooks, I. M., Browse, J., Burrows,
S. M., Carslaw, K. S., Huffman, J. A., Judd, C., Kilthau, W. P., Mason, R. H., McFiggans, G.,
Miller, L. A., Nájera, J. J., Polishchuk, E., Rae, S., Schiller, C. L., Si, M., Temprado, J. V.,
Whale, T. F., Wong, J. P. S., Wurl, O., Yakobi-Hancock, J. D., Abbatt, J. P. D., Aller, J. Y.,
Bertram, A. K., Knopf, D. A. and Murray, B. J.: A marine biogenic source of atmospheric ice-
nucleating particles, Nature, 525(7568), 234–238, doi:10.1038/nature14986, 2015.

Wright, T. P. and Petters, M. D.: The role of time in heterogeneous freezing nucleation, J.
Geophys. Res. Atmos., 118(9), 3731–3743, doi:10.1002/jgrd.50365, 2013.

Wright, T. P., Petters, M. D., Hader, J. D., Morton, T. and Holder, A. L.: Minimal cooling rate
dependence of ice nuclei activity in the immersion mode, J. Geophys. Res. Atmos., 118(18),
10,535-10,543, doi:10.1002/jgrd.50810, 2013.

Yin, Y., Wurzler, S., Levin, Z. and Reisin, T. G.: Interactions of mineral dust particles and
clouds: Effects on precipitation and cloud optical properties, J. Geophys. Res. Atmos., 107(23),
1–14, doi:10.1029/2001JD001544, 2002.

Zobrist, B., Marcolli, C., Peter, T. and Koop, T.: Heterogeneous Ice Nucleation in Aqueous
Solutions: the Role of Water Activity, J. Phys. Chem. A, 112(17), 3965–3975,
doi:10.1021/jp7112208, 2008.

Zolles, T., Burkart, J., Häusler, T., Pummer, B., Hitzenberger, R. and Grothe, H.: Identification
of Ice Nucleation Active Sites on Feldspar Dust Particles, J. Phys. Chem. A, 119(11), 2692–
2700, doi:10.1021/jp509839x, 2015.